# MKP1 promotes nonalcoholic steatohepatitis by suppressing AMPK activity through LKB1 nuclear retention

Bin Qiu [1,2], Ahmed Lawan [3], Chrysovalantou E. Xirouchaki[4,5], Jae-Sung Yi[1,2], Marie Robert[6], Lei Zhang[1,2], Wendy Brown[7], Carlos Fernández-Hernando [2,6,8,9], Xiaoyong Yang [2,9], Tony Tiganis [4,5,9] & Anton M. Bennett [1,2,8,9] ✉

Nonalcoholic steatohepatitis (NASH) is triggered by hepatocyte death through activation of caspase 6, as a result of decreased adenosine monophosphate (AMP)-activated protein kinase-alpha (AMPKα) activity. Increased hepatocellular death promotes inflammation which drives hepatic fibrosis. We show that the nuclear-localized mitogen-activated protein kinase (MAPK) phosphatase-1 (MKP1) is upregulated in NASH patients and in NASH diet fed male mice. The focus of this work is to investigate whether and how MKP1 is involved in the development of NASH. Under NASH conditions increased oxidative stress, induces MKP1 expression leading to nuclear p38 MAPK dephosphorylation and decreases liver kinase B1 (LKB1) phosphorylation at a site required to promote LKB1 nuclear exit. Hepatic deletion of MKP1 in NASH diet fed male mice releases nuclear LKB1 into the cytoplasm to activate AMPKα and prevents hepatocellular death, inflammation and NASH. Hence, nuclear-localized MKP1-p38 MAPK-LKB1 signaling is required to suppress AMPKα which triggers hepatocyte death and the development of NASH.

Nonalcoholic fatty liver disease (NAFLD) progresses from nonalcoholic fatty liver (NAFL), which is defined as fat accumulation in the liver without either liver cell death or inflammation to nonalcoholic steatohepatitis (NASH), a progressive subset of NAFLD, where the initiation of liver cell death triggers inflammation that leads to fibrosis[1–4]. NASH occurs in ~20–30% of NAFLD patients and is characterized by chronic lipid accumulation, liver damage (hepatocyte ballooning, apoptosis, and focal necrosis) and lobular and/or portal inflammation involving the recruitment and activation of immune cells[3,5]. Chronic hepatocellular death triggers a reparative response which ultimately causes fibrosis. NASH represents the leading cause of liver transplantation and hepatocellular carcinoma[5]. NAFLD is estimated to occur in ~25% of the population and the prevalence of NASH is estimated to be ~20%[6]. The mechanisms governing the transition from NAFL to NASH are complex and ~15% of individuals with NASH progress to cirrhosis and these individuals have increased susceptibility to develop hepatocellular carcinoma[3,5]. Furthermore, NASH is highly associated with cardiovascular disease which is a major cause of death in these patients[7]. There is currently no effective pharmacological therapy for NASH, and efforts to control complications arising from the condition are far from satisfactory[3]. Therefore, understanding the underlying mechanisms

[1]Yale University School of Medicine, Department of Pharmacology, 333 Cedar Street, New Haven, CT 06520, USA. [2]Yale University School of Medicine, Yale Center of Molecular and Systems Metabolism, New Haven, CT 06520, USA. [3]University of Alabama, Department of Biological Sciences, 301 Sparkman Drive, Huntsville, AL 35899, USA. [4]Monash Biomedicine Discovery Institute, Monash University, Clayton, VIC 3800, Australia. [5]Department of Biochemistry and Molecular Biology, Monash University, Clayton, VIC 3800, Australia. [6]Yale University School of Medicine, Department of Pathology, 300 Cedar Street, New Haven, CT 06520, USA. [7]Monash University Department of Surgery, Alfred Hospital, Melbourne, Victoria 3004, Australia. [8]Yale University School of Medicine, Vascular Biology and Therapeutics Program, New Haven, CT 06520, USA. [9]Department of Comparative Medicine, Yale University School of Medicine, New Haven, CT, USA. ✉e-mail: anton.bennett@yale.edu

that govern the development of NASH will be instrumental in identifying new modalities for therapeutic intervention.

The dysregulation of mitogen-activated protein kinase (MAPK) signaling has been implicated in playing a role in diseases such as metabolic disorders[8–10]. There are three major MAPKs, the extracellular signal-regulated kinases 1 and 2 (ERK1/2), c-Jun NH$_2$-terminal kinases (JNK) and p38α/β MAPK (p38 MAPK)[11–13]. It has been suggested that enhanced p38 MAPK and JNK activities promote the development of NAFL, with JNK being a prominent driver of this process[8,9,14–17]. However, for p38 MAPK it is less clear whether it promotes or antagonizes NAFL as conflicting effects have been reported[9]. Liver-specific p38 MAPK knockout mice exhibited exacerbated steatosis when fed a high-fat diet (HFD) indicating that p38 MAPK protects against NAFL[18]. Further it has been shown that in the livers of obese mice that p38 MAPK activity is reduced[19]. These results would indicate that hepatic MAPKs play a more complex role in the development of liver disease than is currently appreciated. Despite intensive investigation into the role of the MAPKs in obesity and type II diabetes[9,20–22] it remains unclear whether the MAPKs, and more specifically hepatic MAPKs, are involved in NASH. Given the irreversible nature of advanced NASH with severe fibrosis it is of significance that identifying the mechanisms of NASH be thoroughly defined.

MAPK phosphatase-1 (MKP1) is a nuclear-localized phosphatase, that inactivates the nuclear pool of p38 MAPK, JNK and ERK1/2 in all cell types where it is expressed including the liver[23]. MKP1 was identified as an immediate-early gene that is upregulated in response to oxidative stress and liver regeneration[24,25]. We have demonstrated that MKP1 plays an important role in the maintenance of MAPK signaling in metabolism[26–30], and is upregulated in liver diseases associated with metabolic syndrome[9,26,27]. MKP1 has been shown to exhibit gene polymorphisms that are associated with obesity-related metabolic complications in severely obese patients[31] and MKP1 is upregulated in circulating mononuclear cells, subcutaneous adipose tissue and skeletal muscle in obese non-diabetic patients[32,33]. These correlative observations argue that in humans increased MKP1 expression is associated with metabolic disease. Consistent with this, we showed that mice lacking hepatic MKP1, which results in increased p38 MAPK and JNK activity, are resistant to NAFL[26]. Lipotoxicity and oxidative stress are considered amongst the initial triggers of NASH[34–36]. During the progression of NAFL, hepatic mitochondria are structurally and molecularly altered due to lipotoxicity, which contributes to hepatocellular death[37], a cardinal feature of NASH. Given that MKP1 functions as a stress-responsive gene its actions in the liver under chronic conditions of lipotoxic stress may become dysregulated leading to hepatocellular death and subsequently NASH. Furthermore, it is unclear whether dysregulation of the MAPKs themselves is involved in the development of NASH. Thus, one of the goals of this study was to investigate the role of MKP1, and thus the MAPKs, in the development of NASH.

Adenosine monophosphate (AMP)-activated protein kinase (AMPK) is a key metabolic regulator that senses energy status and controls energy expenditure and storage[38,39]. When cellular energy levels are depleted, and the AMP:ATP ratio rises, AMPK is activated via allosteric binding of AMP to the AMPKα subunit, which induces a conformational change in the complex, facilitating the ability of AMPKα to serve as a substrate for upstream kinases[40]. It has been reported that the activity of AMPKα is suppressed in animal models of NASH[41–43]. Moreover, liver-specific constitutive AMPKα activation protects against diet-induced obesity, hepatic steatosis, and liver fibrosis[43,44]. AMPKα activity is decreased in diabetic patients with NAFLD despite there being paradoxically reduced ATP concentrations[45]. Consistent with this it has been shown that inactivation of AMPK does not further promote the development of NAFLD[43]. These observations suggest that AMPK downregulation and fatty liver development are uncoupled. However, the mechanism of

how hepatic AMPK activity is repressed despite reduced liver ATP concentrations in NAFLD/NASH remains unclear[42,43,45]. The binding of AMP to the regulatory subunit of AMPK induces a conformational change that allows for its phosphorylation and activation by the liver kinase B1 (LKB1). LKB1 contains a nuclear localization signal and its ability to phosphorylate AMPKα relies upon its nuclear export to the cytosol where the vast majority of AMPK resides[46]. Although LKB1 activity appears to be involved with its nuclear to cytosolic shuttling[47,48] how this is regulated is not fully understood and whether such pathways are disrupted in NAFLD/NASH is unknown. Although, a repression of AMPK in NAFLD/NASH correlates with a state of excess nutrients how LKB1 is regulated under these conditions to restrict and thus lower AMPK activity is unclear. Furthermore, it has been proposed that hepatic AMPK downregulation promotes hepatocellular death which triggers inflammation and hepatic fibrosis[42]. Interestingly, it has been reported that AMPKα2 can phosphorylate MKP1 resulting in MKP1 ubiquitination-dependent degradation in adipocytes and hepatocytes[49]. However, the spatial constraints between MKP1 nuclear and AMPK cytosolic localization suggests other operative mechanisms of MKP1 regulation by AMPK may exist. Whether the depression of AMPK in NASH alters MKP1-mediated MAPK signaling or whether MKP1 in response to metabolic dysfunction promotes AMPK downregulation in the NAFL to NASH transition is unclear. Thus, understanding how AMPK downregulation occurs and its subsequent effects on hepatocellular death and inflammation could reveal important insight into the mechanisms of NASH.

In this study, we determined the role of MKP1 in NASH development and uncover the mechanism by which MKP1 depresses AMPK to promote NASH. The results show that MKP1 is upregulated in mice fed NASH-inducing diets and in patients with NASH. Hepatocyte-specific deletion of MKP1 in mice prevents the pathogenesis of NASH. Hepatic MKP1 localizes to the nucleus and its upregulation in NAFL as a result of oxidative stress promotes hepatocyte death which triggers an inflammatory response and subsequently fibrosis. Mechanistically, MKP1 dephosphorylates p38 MAPK in the nucleus which reduces LKB1 phosphorylation on Ser428, a site required for LKB1 nuclear exit, thus limiting AMPKα activation to negatively regulate caspase 6-mediated apoptosis. These results provide the first insights into the MKP1/MAPK regulatory network in NASH.

## Results

### Upregulation of MKP1 in human NASH patients and requirement for development of NASH in mice

Hepatic steatosis is a major risk factor for the development of NASH in humans[3,5]. The accumulation of free fatty acids (FFA) in the liver as a result of dysfunctional FFA metabolism overwhelms the physiological regulators of redox homeostasis in the hepatocyte[50]. Once this occurs, oxidative stress ensues which causes hepatocyte death and liver injury[37,50]. Hepatocyte death initiates the repair process that promotes fibrosis[37,51]. To study the involvement of MKP1 in hepatocyte death and thus progression from NAFL to NASH we first determined by quantitative real-time PCR the expression of *DUSP1* mRNA levels in core liver biopsies from obese non-steatotic (BMI; 36–61, NAS = 0), obese steatotic (BMI; 36–61, NAS = 1–2) and obese NASH (BMI; 36–61; NAS > 5, fibrosis score = 1–2) patients[52]. We found that *DUSP1* mRNA levels in core liver biopsies were significantly elevated by ~2-fold in obese steatotic (BMI; 36–61, NAS = 1–2) and obese NASH (BMI; 36–61; NAS > 5, fibrosis score = 1–2) patients as compared with obese non-steatotic (BMI; 36–61, NAS = 0) patients (Fig. 1a). These results demonstrate that steatosis rather than obesity per se is likely responsible for the induction of hepatic *DUSP1* mRNA expression. Furthermore, elevated hepatic *DUSP1* mRNA expression levels persist in obese NASH patients. These observations demonstrate that in patients *DUSP1* mRNA levels are elevated in NAFL and persist during the transition from NAFL to NASH.

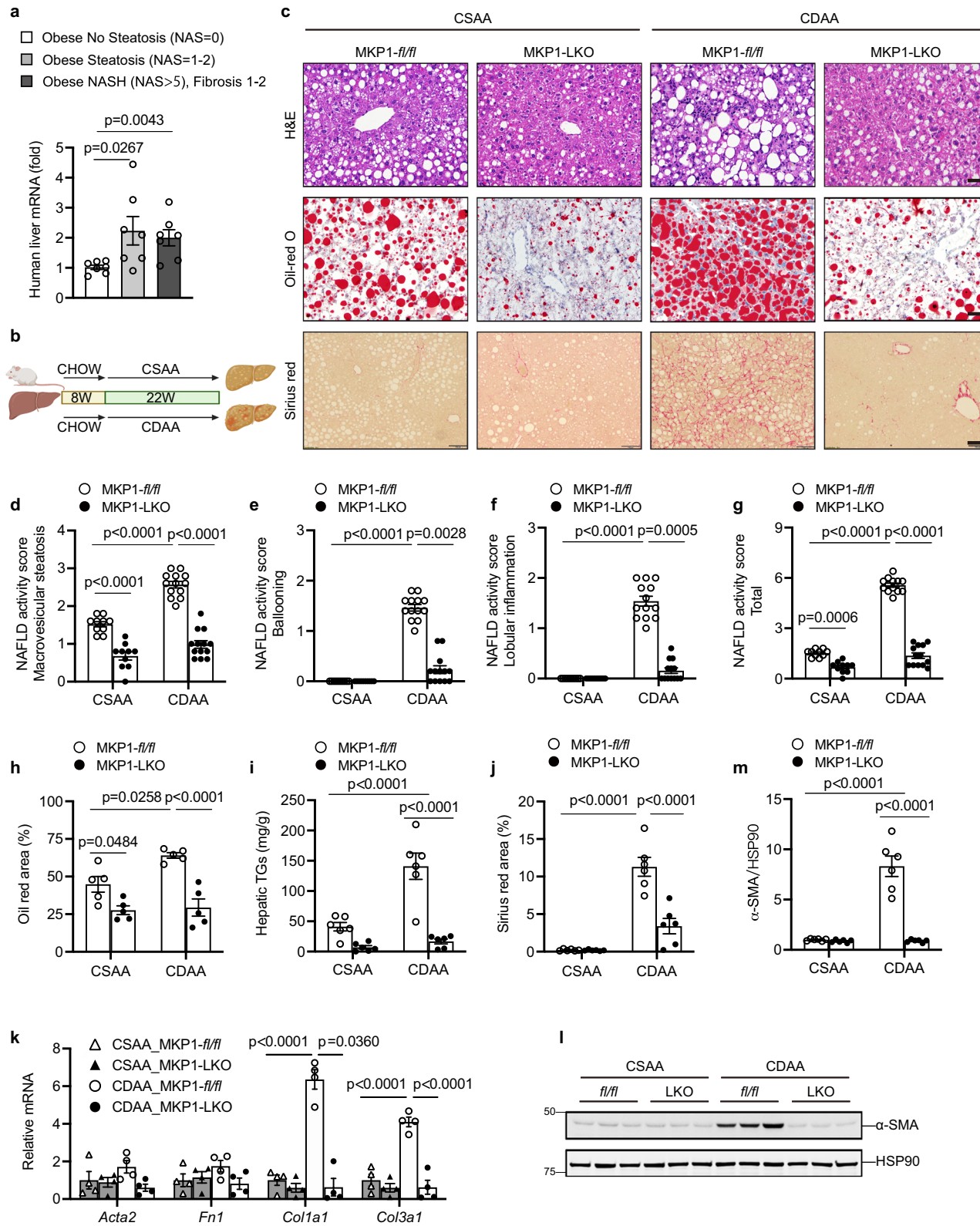

To study the role of MKP1 in NASH we used the choline-deficient L-amino acid-defined (CDAA) diet[53–56]. The CDAA diet induces inflammation and liver fibrosis that approximates to the attributes of NASH in humans[35,53]. Feeding mice with a CDAA diet results in impairment of β-oxidation and production of very low-density lipoprotein particles, resulting in hepatic fat accumulation, liver cell death, oxidative stress, inflammation, and fibrosis. CDAA diet feeding of *Mkp1^flox/flox^* (*Mkp1^fl/fl^*)

mice results in weight gain and the control choline-sufficient L-amino acid-defined (CSAA) diet also induces weight gain to levels equivalent with CDAA-fed mice without promoting NASH (Supplemental Fig. 1a). In contrast, CDAA-fed mice develop NASH as defined by the development of hepatic fibrosis, increased inflammation and hepatocyte ballooning and apoptosis[35]. To establish that MKP1 is specifically upregulated in hepatocytes under conditions that promote liver

**Fig. 1 | Hepatic MKP1 is upregulated in NASH and required for NASH development. a** Quantitative real-time PCR (qPCR) analysis of *DUSP1* in human liver core biopsies from obese patients without steatosis (*n* = 7; NAS = 0), NAFL (*n* = 7; NAS = 1–2) or NASH and fibrosis (*n* = 7; NAS > 5, fibrosis score = 1–2). **b** Schematic representation of CSAA/CDAA diet feeding protocol. **c**–**m** *Mkp1*$^{fl/fl}$ and MKP1-LKO mice were fed with either CSAA or CDAA diet for 22 weeks. **c** Histological H&E staining (scale bar = 50 μm), Oil red-O staining (scale bar = 50 μm) and Sirius red staining (scale bar = 100 μm) from liver sections. **d**–**g** NAFLD activity score for (**d**) macrovesicular steatosis, (**e**) ballooning, (**f**) lobular inflammation and (**g**) total NAFLD score derived from H&E staining in (**c**). Data in (**d**–**g**) represent the mean ± SEM derived from 10 mice for CSAA diet per genotype and 13 mice for CDAA diet per genotype. **h** Quantification of Oil red O-stained areas from Oil red-O

staining in (**c**). Data represent the mean ± SEM derived from 5 mice per group. **i** Triglyceride (TG) content in livers. Data represent the mean ± SEM derived from 5 mice per group. **j** Quantitation of Sirius red-stained areas in (**c**). Data represent the mean ± SEM derived from 6 mice per group. **k** mRNA expression of fibrotic genes in livers. Data represent the mean ± SEM derived from 4 mice per group. **l** Immunoblot of α-smooth muscle actin (α-SMA) protein expression in livers. **m** Densitometry of immunoblot from (**l**). Data represent the mean ± SEM from 6 mice per group. *p*-values shown in (**a**) was determined by one-way ANOVA, shown in (**d**), (**g**-**k**) and (**m**) were determined by two-way ANOVA, shown in (**e**) and (**f**) were determined by Kruskal–Wallis test. Key: *fl/fl*, *Mkp1*$^{fl/fl}$ mice; LKO, MKP1-LKO mice; 8 W, 8 weeks; 22 W, 22 weeks.

fibrosis we isolated hepatocytes from CSAA and CDAA fed *Mkp1*$^{fl/fl}$ mice and examined MKP1 expression. We found that MKP1 protein expression is markedly upregulated in hepatocytes derived from CDAA fed *Mkp1*$^{fl/fl}$ mice as compared with CSAA fed *Mkp1*$^{fl/fl}$ mice (Supplemental Fig. 1b, c). The upregulation of MKP1 occurred concomitant with the downregulation of phosphorylated p38 MAPK and JNK but not ERK in hepatocytes derived from CDAA-fed mice as compared with CSAA-fed mice (Supplemental Fig. 1b, c). Additionally, when hepatocytes derived from CDAA fed *Mkp1*$^{fl/fl}$ mice were compared with those from hepatocyte-specific *Mkp1*-deleted mice (MKP1-LKO) mice fed a CDAA diet the levels of p38 MAPK and JNK phosphorylation, but not ERK phosphorylation were enhanced (Supplemental Fig. 1b, c). Thus, consistent with MKP1 being a major phosphatase that inactivates p38 MAPK and JNK[26], MKP1 is upregulated concomitant with the downregulation of these MAPKs in hepatocytes of CDAA-fed mice that progress to liver fibrosis.

To test whether MKP1 is involved in NASH we utilized MKP1-LKO mice which we have shown are protected from the development of hepatic steatosis when fed a high fat diet[26]. Male *Mkp1*$^{fl/fl}$ and MKP1-LKO mice were fed either a CSAA or CDAA diet for 22 weeks (Fig. 1b). Both CSAA and CDAA-fed *Mkp1*$^{fl/fl}$ and MKP1-LKO mice gained weight at comparable levels (Supplemental Fig. 2a, b). After 22 weeks of CDAA feeding, neither *Mkp1*$^{fl/fl}$ nor MKP1-LKO mice exhibited significant differences in liver to body weight or body composition (Supplemental Fig. 2c–i) indicating that CDAA diet fed mice as compared with CSAA control fed mice were not globally compromised metabolically. Hepatic steatosis developed in *Mkp1*$^{fl/fl}$ mice both under 22 weeks of CSAA and CDAA diet feeding, albeit to a higher extent in CDAA diet fed animals (Fig. 1c). Compared with *Mkp1*$^{fl/fl}$ mice, MKP1-LKO mice were resistant to the development of hepatic steatosis when fed either a CSAA or CDAA diet as shown by markedly reduced macrovesicular steatosis (Fig. 1d), microvesicular steatosis (Supplemental Fig. 2j), hepatocyte ballooning (Fig. 1e) and lobular inflammation (Fig. 1f). The total assessed NAFL activity score was significantly reduced in both CSAA and CDAA-fed MKP1-LKO mice as compared with *Mkp1*$^{fl/fl}$ mice although this difference was more profound in CDAA-fed animals (Fig. 1g). Consistent with this, quantitative assessment of the Oil Red O-stained area and levels of hepatic triglycerides demonstrated significant reduction in neutral lipid accumulation in the livers of CDAA-fed MKP1-LKO mice as compared with *Mkp1*$^{fl/fl}$ mice (Fig. 1h, i).

Next, we examined whether MKP1 affected the development of liver fibrosis in CDAA-fed MKP1-LKO mice. A key hallmark of fibrosis is the accumulation of collagen deposition which we determined by Sirius red staining. Quantitation of the Sirius red-stained areas in *Mkp1*$^{fl/fl}$ mice fed a CDAA diet for 22-weeks showed an extensive area of Sirius red stained regions (Fig. 1c, j). In contrast, MKP1-LKO mice fed a CDAA diet were devoid of Sirius red staining in the liver to levels comparable to that of mice fed the control CSAA diet (Fig. 1c, j). The mRNA expression of fibrotic genes, α-smooth muscle actin (*Acta2*), fibronectin (*Fn1*), collagens (*Col1a1* and *Col3a1*) and the expression of α-smooth muscle actin protein levels were readily detected in *Mkp1*$^{fl/fl}$ mice fed a CDAA diet (Fig. 1c, k–m). However, CDAA-fed MKP1-LKO

mice were completely protected from the activation of these fibrotic genes and the expression of α-smooth muscle actin protein (Fig. 1c, k–m). Importantly, liver fibrosis was undetectable under the control CSAA diet as shown in either *Mkp1*$^{fl/fl}$ or MKP1-LKO mice (Fig. 1c, k–m). These data demonstrate that hepatic MKP1 participates in the development of NASH.

## Reduced apoptosis and inflammation in NASH diet-fed MKP1-LKO mice

A cardinal feature of NASH is the onset of hepatocellular death[35,42]. During the progression of NAFLD, hepatic mitochondria are structurally and molecularly altered due to increased lipotoxic stress which produces mitochondrial $O_2^-$ and $H_2O_2$, which contributes to hepatocellular death[37,50]. We found a significant amount of apoptosis in livers from CDAA diet-fed *Mkp1*$^{fl/fl}$ mice as shown by increased terminal deoxynucleotidyl transferase dUTP nick end-labeling (TUNEL) positive cells (Fig. 2a, b). Further the mechanism of hepatocellular death was ascribed to the activation of apoptosis as demonstrated by the increased levels of cleaved caspase 3 in CDAA diet-fed *Mkp1*$^{fl/fl}$ mice (Fig. 2c). TUNEL-positive hepatocytes and cleaved caspase 3 expression levels were undetectable in the livers derived from CSAA-diet-fed *Mkp1*$^{fl/fl}$ and MKP1-LKO mice (Fig. 2a–c). In contrast, CDAA-fed MKP1-LKO mice were devoid of hepatic TUNEL-positive hepatocytes (Fig. 2a, b) and expressed significantly reduced cleaved caspase 3 levels in the liver (Fig. 2c). These results demonstrate that mice lacking hepatic MKP1 are protected from undergoing apoptosis when fed a CDAA diet and thus reveal a critical role for MKP1 in controlling hepatocellular death in response to a NASH-inducing diet.

A requisite feature of NASH that accompanies hepatocellular death is the recruitment of immune cells that contribute to the repair process[57–59]. *Mkp1*$^{fl/fl}$ mice fed a CDAA diet exhibited a marked increase in the level of immune infiltrates whereas, CDAA diet fed MKP1-LKO mice were devoid of inflammatory infiltrates (Supplemental Fig. 3a). To investigate this further liver tissue sections were subjected to CD45 staining and CD45 mRNA expression analyses. As expected CDAA-fed *Mkp1*$^{fl/fl}$ mice showed a significantly increased percentage of CD45-positive cells (Fig. 3a, b and Supplemental Fig. 3b). In contrast, MKP1-LKO mice fed a CDAA diet expressed a percentage of CD45-positive cells equivalent to that of control CSAA-fed *Mkp1*$^{fl/fl}$ and MKP1-LKO mice (Fig. 3a, b and Supplemental Fig. 3b). These data were corroborated by the observation that CD45 mRNA expression in the livers of CDAA-fed MKP1-LKO mice were equivalent to that of CSAA fed control *Mkp1*$^{fl/fl}$ mice (Fig. 3c). We also observed similar results when assessing macrophage infiltration into the livers of CDAA-fed mice. We detected significant increases in the percentage of CD68-positive cells (Fig. 3d, e and Supplemental Fig. 3c) and increased CD68 mRNA levels in CDAA-fed control *Mkp1*$^{fl/fl}$ mice (Fig. 3f). However, in MKP1-LKO mice fed a CDAA diet CD68-positive cells in the liver were at levels equivalent to that of CSAA-fed *Mkp1*$^{fl/fl}$ mice (Fig. 3d, e and Supplemental Fig. 3c). This was supported by hepatic CD68 mRNA expression levels in MKP1-LKO mice fed a CDAA diet that was similar to CSAA-fed *Mkp1*$^{fl/fl}$ mice (Fig. 3f). Collectively, these data demonstrate that MKP1-deficiency in

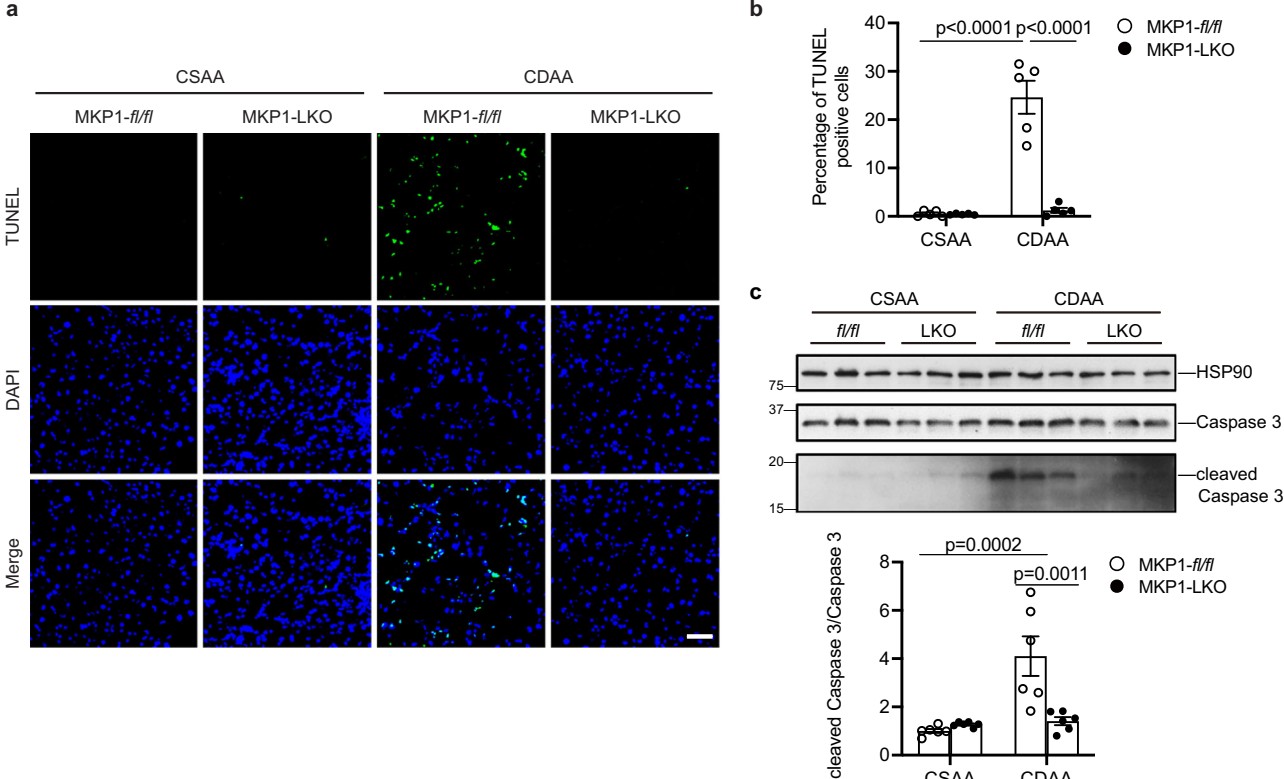

**Fig. 2 | Loss of hepatic MKP1 prevents NASH diet-induced liver apoptosis.**
*Mkp1^{fl/fl}* and MKP1-LKO mice were fed with CSAA or CDAA diet for 22 weeks.
**a** TUNEL staining was applied for apoptosis analysis in liver sections. Scale
bar = 50 µm. **b** Quantitation of percentage TUNEL-positive hepatocytes (%). Data
represent the mean ± SEM from 5 mice per genotype. **c** Immunoblot of cleaved

caspase 3 and caspase 3 in livers. Lower panel represents the densitometry of
immunoblots from cleaved caspase 3/caspase 3. Data represent the mean ± SEM
derived from 6 mice per group. *p*-values were determined by two-way ANOVA.
Key: *fl/fl, Mkp1^{fl/fl}* mice; LKO, MKP1-LKO mice.

hepatocytes of mice fed a NASH-inducing diet ameliorates cell death
and the associated infiltration of immune cells.

**Increased MAPK and AMPK activity in MKP1-LKO mice**
We previously demonstrated that loss of hepatic MKP1 results in
increased activities of nuclear p38 MAPK and JNK in mice fed a HFD[26,30].
To test whether MKP1 also serves as a major negative regulator of the
nuclear p38 MAPK and JNK pools in the livers of mice fed the CDAA diet
we examined the phosphorylation levels of these MAPKs in both the
nucleus and cytosol. CDAA diet-fed MKP1-LKO mice exhibited
increased levels of p38 MAPK in both the nucleus and cytosol, how-
ever, the hyperactivation of p38 MAPK was substantively more pro-
found in the nucleus as compared with *Mkp1^{fl/fl}* mice (Fig. 4a–d).
Similarly, we observed that JNK was hyperactivated in the nucleus and
to a much a lesser extent than in the cytosol of livers derived from
CDAA fed MKP1-LKO mice as compared with *Mkp1^{fl/fl}* controls
(Fig. 4a–d). Despite both p38 MAPK and JNK exhibiting hyperactivation
in MKP1-LKO mice fed a CDAA diet ERK activity remained unaffected
(Fig. 4a–d). These results show that the resistance to the development
of NASH in MKP1-LKO mice fed a CDAA-diet correlated with hyper-
activation of both hepatic p38 MAPK and JNK.

To explore the mechanism by which MKP1 promotes hepatocel-
lular death we hypothesized that the reduced activation of caspase 3 in
hepatocytes of MKP1-LKO mice (Fig. 2) could be due to elevated
AMPKα levels. Reduced AMPKα renders hepatocytes susceptible to
death through a mechanism involving dephosphorylation of pro-
caspase 6 together with Bid, caspase 9, 3 and 7 that initiates a feed-
forward loop that leads to hepatocyte death and the promotion
NASH[41,42]. AMPK is a key metabolic regulator that senses energy status
and controls energy expenditure and storage[38,39] and its activity is

suppressed in obesity and NASH[41–43]. Moreover, liver-specific AMPKα
activation protects against diet-induced obesity, hepatic steatosis[43,44]
and liver fibrosis[42]. Therefore, we assessed the status of AMPK in
CDAA-fed mice. Consistent with previous studies, we found that
hepatic AMPKα activation was suppressed in CDAA-fed *Mkp1^{fl/fl}* as
compared to CSAA-fed *Mkp1^{fl/fl}* mice (Fig. 4e, f). In contrast, the livers of
MKP1-LKO mice fed a CSAA or CDAA diet exhibited significantly
enhanced levels of AMPKα as compared with *Mkp1^{fl/fl}* mice (Fig. 4e, f).
Under all genotypes and conditions, the phosphorylation of AMPKβ1
remained unaffected (Fig. 4e, g). Consistent with the observation that
MKP1 negatively regulates AMPKα the AMPKα substrate involved in
autophagy, ULK-1, showed decreased levels of phosphorylation in the
livers of CDDA-fed *Mkp1^{fl/fl}* mice but this was enhanced in MKP1-LKO
mice fed a CSAA or CDAA diet (Fig. 4e, h). These results demonstrate
that loss of hepatocyte MKP1 leads to elevated AMPKα activity and the
prevention of NASH.

Next, we tested whether hepatic MKP1 is involved in the control of
AMPKα-mediated phosphorylation of the proapoptotic caspase 6
which results in its inactivation[42]. First, we found that caspase 6
phosphorylation at Ser257 by AMPK was increased in CSAA and CDAA-
fed MKP1-LKO mice as compared with *Mkp1^{fl/fl}* mice (Fig. 4i, j). Con-
sistent with reduced AMPKα activity *Mkp1^{fl/fl}* mice fed a CDAA diet
exhibited increased levels of caspase 6 cleavage as measured by the
decreased ratio of phospho-caspase 6/caspase 6 and a concomitant
increase in the expression of cleaved caspase 6, whereas this cleavage
was inhibited in MKP1-LKO mice as compared with *Mkp1^{fl/fl}* mice
(Fig. 4i, k). We performed indirect immunofluorescence for the
expression of cleaved caspase 6 and quantitated the number of posi-
tive hepatocytes from *Mkp1^{fl/fl}* and MKP1-LKO mice fed either a CSAA or
CDAA diet. These results showed a significant reduction in the amount

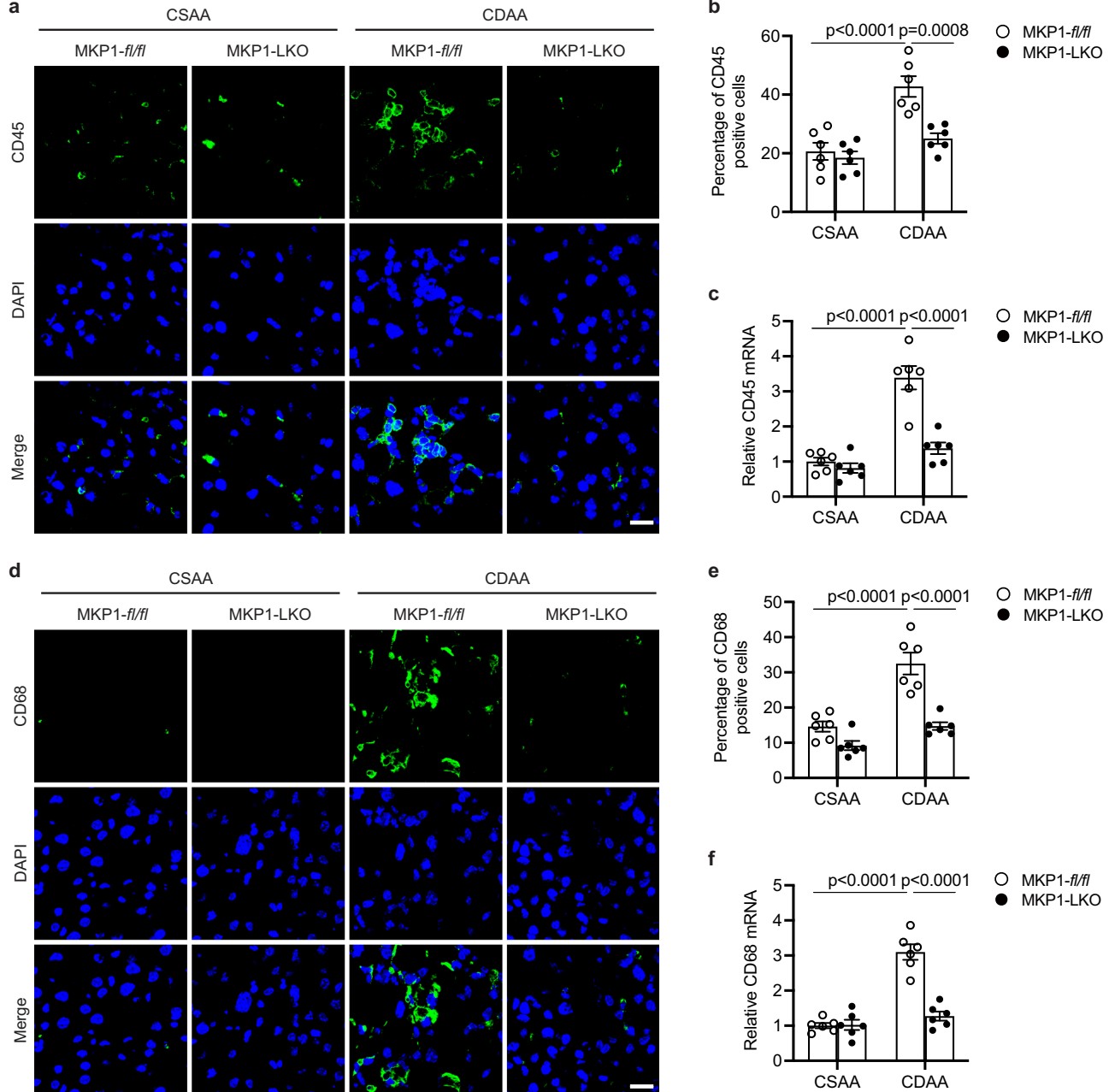

**Fig. 3 | MKP1 is required to promote inflammatory infiltrates in livers of NASH diet fed mice.** *Mkp1$^{fl/fl}$* and MKP1-LKO mice were fed with CSAA or CDAA diet for 22 weeks. **a** and **d** Staining of CD45 and CD68 in liver sections from CSAA or CDAA fed *Mkp1$^{fl/fl}$* and MKP1-LKO mice. Scale bar = 20 μm. **b**, **e** Quantitation of percentage of CD45 and CD68-positive cells in livers from (**a**) and (**d**). **c**, **f** mRNA expression of CD45 and CD68 in livers from CSAA or CDAA fed *Mkp1$^{fl/fl}$* and MKP1-LKO mice. Data represent the mean ± SEM derived from 6 mice per group. *p*-values were determined by two-way ANOVA.

of cleaved caspase 6 in MKP1-LKO mice fed a CDAA diet as compared with *Mkp1$^{fl/fl}$* controls fed the same CDAA diet (Fig. 4l, m and Supplemental Fig. 4). Together with the results presented in Fig. 2, these data demonstrate that increased AMPKα activity in the livers of MKP1-LKO mice fed a CDAA diet is associated with decreased hepatocyte apoptosis through the AMPKα-caspase 6 pathway.

**MKP1 is upregulated by ROS and promotes AMPKα-caspase 6-mediated hepatocellular apoptosis**
To define the mechanistic basis of how hepatic MKP1 regulates the AMPKα-caspase 6 pathway, we utilized HepG2 cells cultured in either a choline-sufficient (CS) or a choline-deficient (CD) medium as a cellular model to mimic the in vivo CSAA/CDAA diet effects[60]. First, we

demonstrated that CD medium-treated HepG2 cells accumulate lipid as measured by BODIPY493/503 immunofluorescence (Supplementary Fig. 5a, b). Consistent with this, when HepG2 cells were cultured in CD medium reactive oxygen species (ROS) production increased as measured by BODIPY 581/591 C11, a lipid peroxidation sensor (Supplementary Fig. 5c, d). Under these conditions, the expression of MKP1 is significantly induced (Fig. 5a, b) concomitant with decreased phosphorylation of p38 MAPK and JNK (Fig. 5a, c–e). The upregulation of ROS and MKP1 protein expression in CD medium-treated HepG2 cells could be inhibited by treating CD-cultured HepG2 cells with either N-acetyl-l-cysteine (NAC) or the mitochondrial-targeted anti-oxidant, MitoTEMPO (Supplementary Fig. 5c–e). These results demonstrated that MKP1 is upregulated in response to oxidative stress in CD-treated

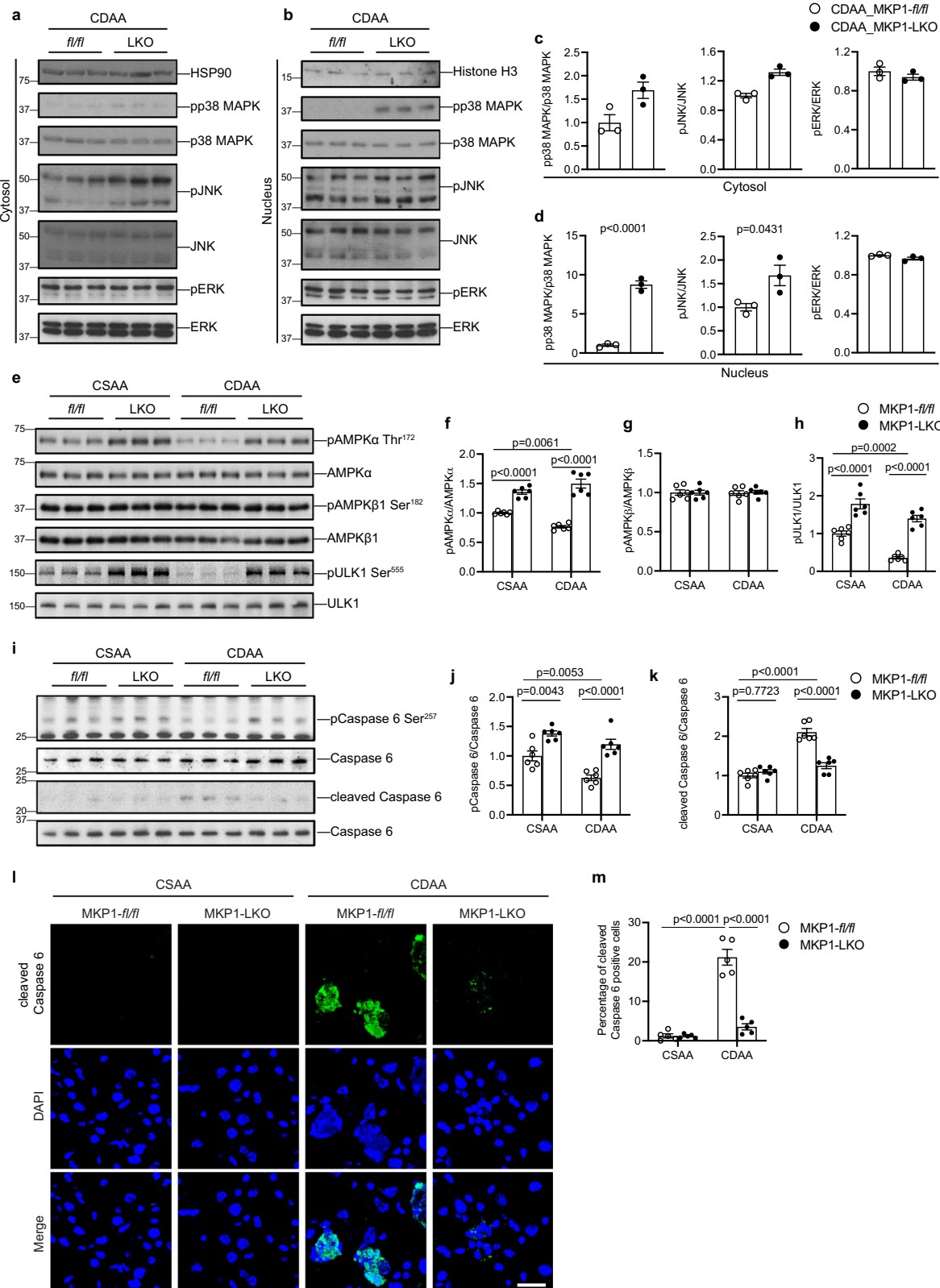

HepG2 cells. When HepG2 cells were treated with CD medium we found that knockdown of MKP1 by siRNA resulted in the expected hyperactivation of p38 MAPK and JNK, but not ERK1/2 (Fig. 5a–e). Importantly, AMPKα activity, phosphorylation of ULK-1 and caspase 6 were elevated significantly, but not AMPKβ phosphorylation, in MKP1 knockdown CD medium-treated HepG2 cells (Fig. 5f–k). Consistent with enhanced AMPKα activity and caspase 6 phosphorylation,

caspase 6 cleavage was completely blocked by MKP1 knockdown in CD medium-treated HepG2 cells (Fig. 5j, l) and this correlated with inhibition of caspase 3 cleavage (Fig. 5j, m). Finally, we measured apoptosis directly in MKP1 knockdown CD medium-treated HepG2 cells. As expected, CD medium-treated HepG2 cells exhibited significantly increased levels of TUNEL-positive cells (Fig. 5n, o). In contrast, MKP1 knockdown in CD medium-treated HepG2 cells were completely

**Fig. 4 | Hepatic MKP1 negatively regulates AMPKα-caspase 6 signaling in NASH diet fed mice. a, b** Immunoblots showing cytoplasmic and nuclear pp38 MAPK, pJNK and pERK1/2 and respective MAPK controls in livers from *Mkp1$^{fl/fl}$* and MKP1-LKO mice fed with CDAA diet for 22 weeks. **c, d** Densitometry of pMAPK/MAPK ratios in the cytosol and nucleus from (**a**) and (**b**), respectively. Data represent the mean ± SEM from 3 mice per genotype. *p*-values shown for cytoplasmic pp38 MAPK/p38 MAPK and pJNK/JNK were determined by a Mann–Whitney test, all other datasets were determined by two-sided student's unpaired *t*-test. **e** Immunoblots showing phospho-AMPKα (Thr172), phospho-AMPKβ1 (Ser182) and phospho-ULK1 (Ser555) with corresponding total protein controls in livers. **f–h** Densitometry of

pAMPKα/AMPKα, pAMPKβ1/AMPKβ1 and pULK1/ULK1 ratios from (**e**). Data represent the mean ± SEM from 6 mice. **i** Immunoblots for phospho-caspase 6 (Ser257), cleaved caspase 6 and caspase 6 in livers. **j, k** Quantitation of the ratio of phospho-caspase 6/caspase 6 and cleaved caspase 6/caspase 6 from (**i**). Data represent the mean ± SEM from 6 mice. **l** Staining of cleaved caspase 6 in liver sections from CSAA or CDAA diet fed *Mkp1$^{fl/fl}$* and MKP1-LKO mice. Scale bar = 25 μm. **m** Quantitation of percentage of cleaved caspase 6-positive cells for (**l**). Data represent the mean ± SEM from 5 mice. *p*-values shown in **f–h, j, k** and **m** were determined by two-way ANOVA. Key: *fl/fl, Mkp1$^{fl/fl}$* mice; LKO, MKP1-LKO mice.

protected from apoptosis and displayed levels of TUNEL staining equivalent to CS medium-treated HepG2 cells (Fig. 5n, o). Collectively, these results show that oxidative stress promotes the upregulation of hepatic MKP1 which contributes to the downregulation of AMPKα and subsequently activation of caspase 6/3-mediated apoptosis.

### MKP1 acts upstream of AMPKα-caspase 6 in a p38 MAPK-dependent manner

To further define the relationship between MKP1 and AMPKα we performed experiments using an AMPKα inhibitor, compound c (CC), to establish whether MKP1 acts upstream of the AMPKα-caspase 6 apoptotic pathway. We found that following MKP1 knockdown in CD-treated HepG2 cells the enhanced level of AMPKα, but not AMPKβ, phosphorylation activity is inhibited by CC (Fig. 6a–c). Next, when MKP1 was knocked down in HepG2 cells cultured in CD medium caspase 6 phosphorylation was increased concomitant with reduced cleavage of both caspase 6 and caspase 3 (Fig. 6d–g). In contrast, CC treatment completely abolished the effects of MKP1 knockdown in HepG2 cells cultured in CD medium on caspase 6 phosphorylation and caspase 6 and 3 cleavage (Fig. 6d–g). As an orthogonal approach we used siRNA knockdown of AMPKα, instead of CC to inhibit AMPKα, in combination with siRNA MKP1 knockdown. These results recapitulated those obtained with CC, whereby siRNA knockdown of MKP1 induced caspase 6 Ser257 phosphorylation and this was rescued when cells were treated with siRNA against AMPKα (Supplemental Fig. 6a–c). Additionally, CD media-treated HepG2 cells in which MKP1 was knocked down led to decreased caspase 6 and 3 cleavage. Both decreased caspase 6 and 3 cleavage were rescued in HepG2 cells in which both MKP1 and AMPKα were knocked down (Supplemental Fig. 6a–e). These results support the assignment of MKP1 as acting upstream of AMPKα. Consistent with the role of MKP1 as an upstream regulator of AMPKα, CC reverted the protective effects conferred by MKP1 knockdown in CD medium-induced apoptosis in HepG2 cells (Fig. 6h, i). Hence, these results strongly support the interpretation that MKP1 acts in a cell autonomous manner upstream of the AMPKα-caspase 3/6 apoptosis pathway.

MKP1-LKO mice fed a CDAA diet are protected from the development of NASH and this correlates with the upregulation of the MKP1 substrates, p38 MAPK and JNK, in the nucleus (Fig. 4a–d). We hypothesized that if MKP1 acts upstream of AMPKα to mediate caspase 6 phosphorylation then it should do so in a manner dependent upon either p38 MAPK and/or JNK. To test this, we assessed the ability of inhibitors of p38 MAPK and JNK to rescue the enhanced activation of AMPKα and caspase 6 phosphorylation in MKP1 siRNA-treated HepG2 cells. We treated HepG2 cells transfected either with a scrambled siRNA or MKP1-specific siRNA in either the presence or absence of the p38 MAPK inhibitor, SB203580 or the JNK inhibitor, SP600125. Under conditions in which MKP1 was knocked down in CD medium treated HepG2 cells both p38 MAPK and JNK were hyperphosphorylated and treatment with either SB203580 or SP600125 blocked p38 MAPK-dependent MAPKAPK2 phosphorylation (Fig. 7a, b) or JNK phosphorylation (Fig. 7g, h), respectively. However, SB203580, but not SP600125, treatment of HepG2 cells in which MKP1 was knocked down in CD medium restored AMPKα and caspase 6 phosphorylation to

levels comparable to DMSO-treated control siRNA transfected HepG2 cells (Fig. 7). Thus, MKP1 acts to dephosphorylate p38 MAPK upstream of AMPKα and caspase 6. Further, these results support the supposition that upregulation of hepatic MKP1 in response to oxidative stress reduces nuclear p38 MAPK activity resulting in the downregulation of AMPKα-mediated phosphorylation of caspase 6 which promotes a positive feed-forward loop activation of caspase 3-mediated hepatocellular apoptosis.

### MKP1-mediated dephosphorylation of p38 MAPK regulates LKB1 nuclear retention

To further confirm the contribution of the effects of MKP1 in the progression of NASH we used the choline-deficient high-fat diet (CDHFD; 35% fat/60% kcal fat) model (Supplemental Fig. 7a)[61]. We again found that MKP1-LKO mice were protected from the development of hepatocellular death, and subsequent inflammation in the livers of MKP1-LKO mice fed a CDHFD for 12 weeks (Supplemental Fig. 7h, i, k, l). CDHFD-fed MKP1-LKO mice were impaired in their ability to develop hepatic fibrosis as measured by the inhibition of fibrogenic genes (Supplemental Fig. 7j). In addition, the livers of MKP1-LKO mice fed a CDHFD showed enhanced AMPKα activation predominately in the cytosol (Fig. 8a–d). Thus, MKP1 is required for the development of NASH in distinct dietary conditions.

MKP1 is nuclear-localized[62] and dephosphorylates the nuclear pool of p38 MAPK under conditions of NASH when upregulated. We reasoned that MKP1-mediated p38 MAPK dephosphorylation regulates AMPKα by controlling its upstream activator, LKB1 which is a nuclear-localized kinase[48,63]. LKB1 is exported to the cytoplasm where it becomes active upon binding to STRADα and MO25 leading to the phosphorylation and activation of AMPKα[47,48]. We performed fractionation experiments in livers derived from *Mkp1$^{fl/fl}$* and MKP1-LKO mice fed a CDHFD. We found that in the livers of MKP1-LKO mice fed a CDHFD, LKB1 exhibited increased expression in the cytosol and concomitant decreased expression in the nucleus (Fig. 8a–d). LKB1 phosphorylation at Serine 428 (Ser428) was analyzed because this site when phosphorylated has been shown to promote LKB1 nuclear to cytosol export[64]. We found that LKB1 Ser428 was increased in the livers of MKP1-LKO mice fed a CDHFD (Fig. 8a–d). To complement these studies, we overexpressed a GFP-tagged vector and a GFP-tagged MKP1 fusion protein (GFP-MKP1) into HepG2 cells and assessed the effect on the sub-cellular localization of co-expressed FLAG-tagged LKB1 (FLAG-LKB1). These experiments revealed that FLAG-LKB1 was restricted entirely to the nucleus in GFP-MKP1 overexpressing cells (Fig. 8e, f) whereas, in GFP alone expressing HepG2 cells FLAG-LKB1 showed a diffuse expression pattern (Fig. 8e, f). Conversely, MKP1 knockdown in HepG2 cells cultured in CD medium resulted in increased expression of endogenous LKB1 in the cytosol with concomitant decreased expression levels in the nucleus (Fig. 9a–c). Similarly, we observed an increase in phosphorylation of the cytosolic AMPKα fraction (Fig. 9a–c). Since we had demonstrated that MKP1 regulates p38 MAPK-mediated AMPKα activity (Fig. 7), we tested whether the sub-cellular localization of LKB1 was p38 MAPK-dependent. The effect of MKP1 knockdown on LKB1 localization to the cytosol was blocked in CD medium treated HepG2 cells in which p38 MAPK activity was

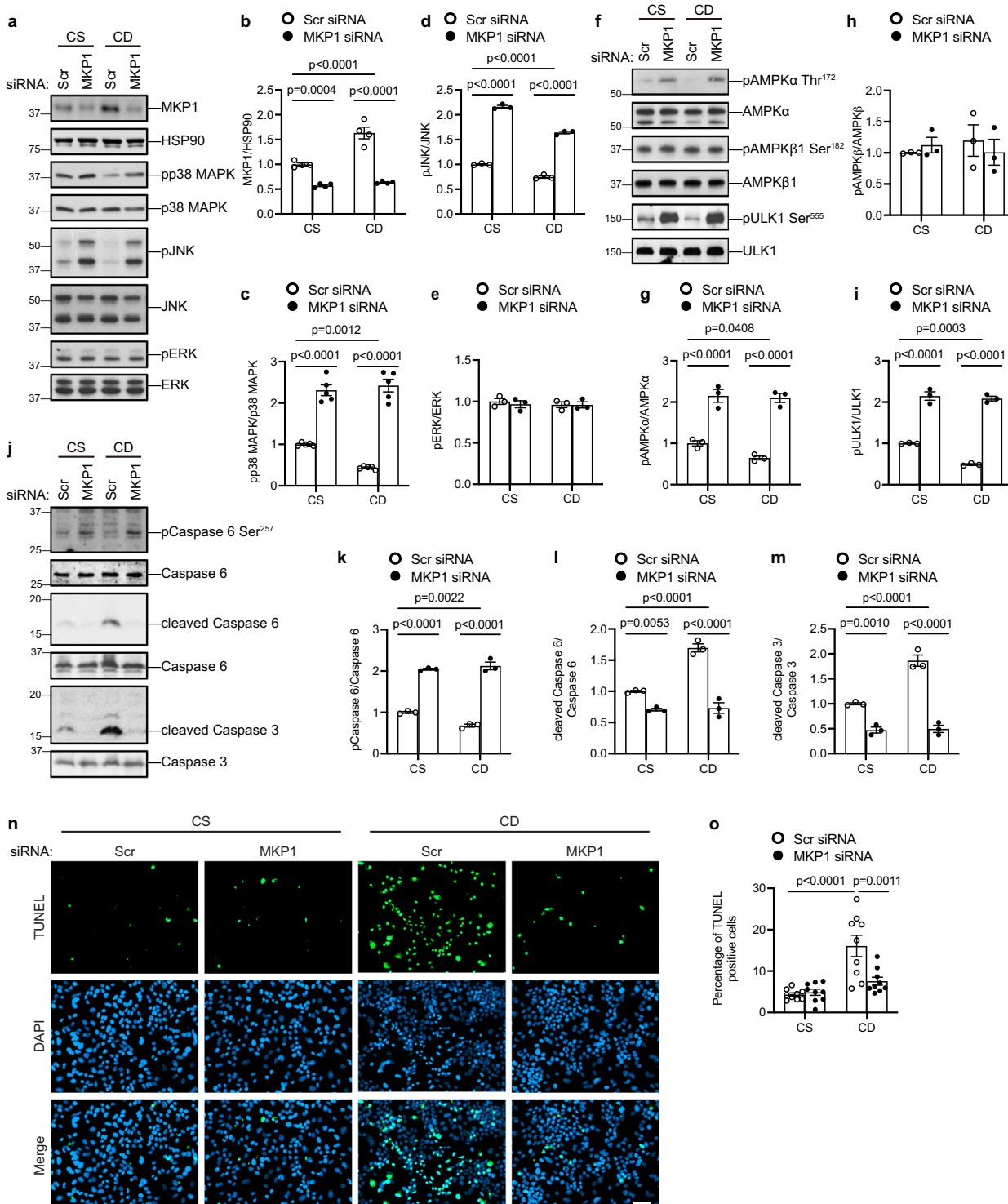

**Fig. 5 | MKP1 negatively regulates AMPKα activity and promotes caspase 6/3-mediated apoptosis.** HepG2 cells were transfected with scrambled siRNA (Scr) or MKP1 siRNA for 24 h followed by 24 h starvation in serum-free medium. Cells were treated with choline-sufficient (CS) or choline-deficient (CD) medium for 4 h (for detecting MKP1, MAPKs, AMPK pathway in **a–i** or 24 h (for detecting apoptosis in **j–o**)). **a** Immunoblots of MKP1 and HSP90 as a loading control or phospho-p38 MAPK, pJNK and pERK with the indicated corresponding MAPK totals. **b–e** Quantitation of immunoblots from (**a**). **f** Immunoblots of phospho-AMPKα (Thr172), phospho-AMPKβ1 (Ser182) and phospho-ULK1 (Ser555) with the indicated corresponding totals. **g–i** Quantitation of immunoblots from (**f**). **j** Immunoblots of phospho-caspase 6 (Ser257), cleaved Caspase 6 and cleaved caspase 3 with the indicated corresponding totals. **k–m** Quantitation of immunoblots from (**j**). **n** Apoptotic cells were stained by TUNEL (green) and nuclei by DAPI (blue). Scale bar = 50 µm. **o** Quantitation of % TUNEL-positive cells from (**n**). Data were collected from 9 fields for each condition, across 3 independent experiments. Key: CS choline-sufficient medium, CD choline-deficient medium. Data represent the mean ± SEM from 3 (**d, e, g–i, k–m, o**), 4 (**b**) or 5 (**c**) independent experiments. *p*-values were determined by two-way ANOVA.

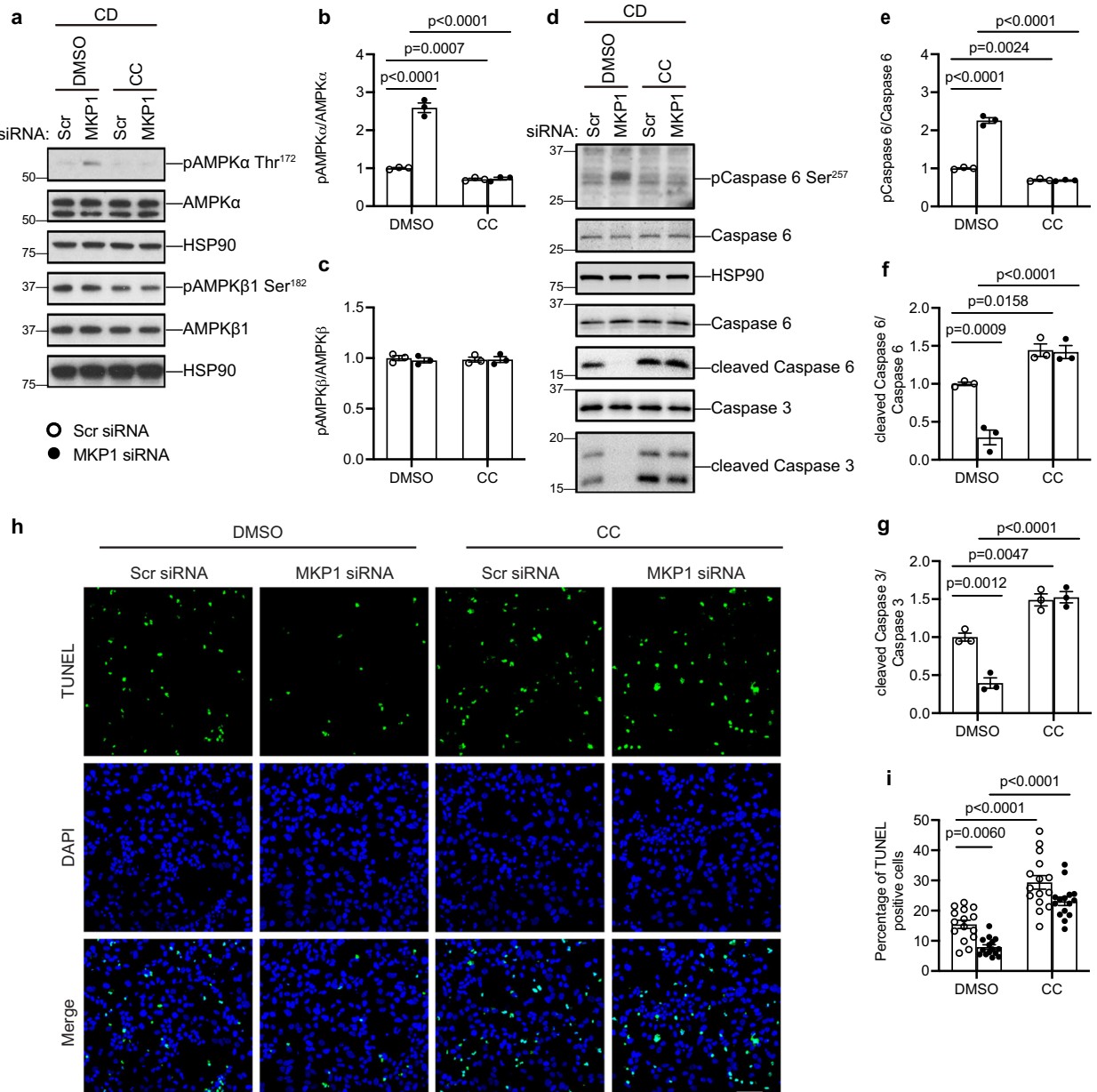

**Fig. 6 | MKP1 acts upstream of the AMPKα-caspase 6 pathway.** HepG2 cells were transfected with scrambled siRNA (Scr) or MKP1 siRNA for 24 h followed by 24 h starvation with serum-free medium. Cells were treated with CD medium in the presence of DMSO or 10 μM compound C (CC) for 4 h (for detecting AMPK pathway in (**a**–**c**)) or 24 h (for detecting apoptosis in **d**–**i**). **a** Immunoblots of phospho-AMPKα (Thr172), phospho-AMPKβ1 (Ser182) with the indicated corresponding AMPK totals. **b**, **c** Quantitation of immunoblots from (**a**). **d** Immunoblots of phospho-caspase 6 (Ser257), cleaved caspase 6 and cleaved caspase 3 with the indicated corresponding totals. **e**–**g** Quantitation of immunoblots from (**d**). **h** Apoptotic cells were stained by TUNEL (green) and nuclei by DAPI (blue). Scale bar = 50 μm. **i** Quantitation of % TUNEL-positive cells from (**h**). Data were collected from 15 fields for each condition, across 3 independent experiments. Key: CD choline-deficient medium, CC compound C. Data represent the mean ± SEM from 3 independent experiments. *p*-values were determined by two-way ANOVA.

inhibited by SB203580 (Fig. 9a–c and Supplemental Fig. 8). Interestingly, we found that in HepG2 cells co-expressing FLAG-LKB1 and HA-tagged p38α MAPK (HA-p38α MAPK) that p38α MAPK forms a complex with LKB1 as assessed by co-immunoprecipitation assays (Fig. 9d, e). Complex formation between p38 MAPK and LKB1 was dependent upon p38 MAPK activity since pre-treatment of HepG2 cells cultured in CD medium with SB203580 resulted in significantly reduced levels of the p38 MAPK/LKB1 complex as compared with controls (Fig. 9d, e). To further confirm the effect of p38 MAPK on LKB1 localization, serum-starved HepG2 cells were treated with the p38 MAPK inhibitor, and the localization of endogenous LKB1 was evaluated. The results showed that LKB1 was distributed both in the cytosol and nucleus, however, the majority of LKB1 accumulated in the nucleus when these cells were

treated with the p38 MAPK inhibitor (Fig. 9f, g). The reduced nuclear pool of active p38 MAPK and impaired ability of p38 MAPK to complex with LKB1 correlates with a decrease in LKB1 nuclear to cytosolic translocation via impairment of LKB1 Ser428 phosphorylation. The resultant reduction in AMPKα activation promotes caspase 6/3-mediated hepatocyte apoptosis.

## Discussion

Understanding the mechanisms of how hepatocellular death is triggered is of paramount importance to our understanding of NASH which remains a poorly understood process[65]. Although positive and negative roles for the stress-activated MAPKs, JNK and p38 MAPK, has been demonstrated for the development of NAFL[8,9,14–17] it has yet to be

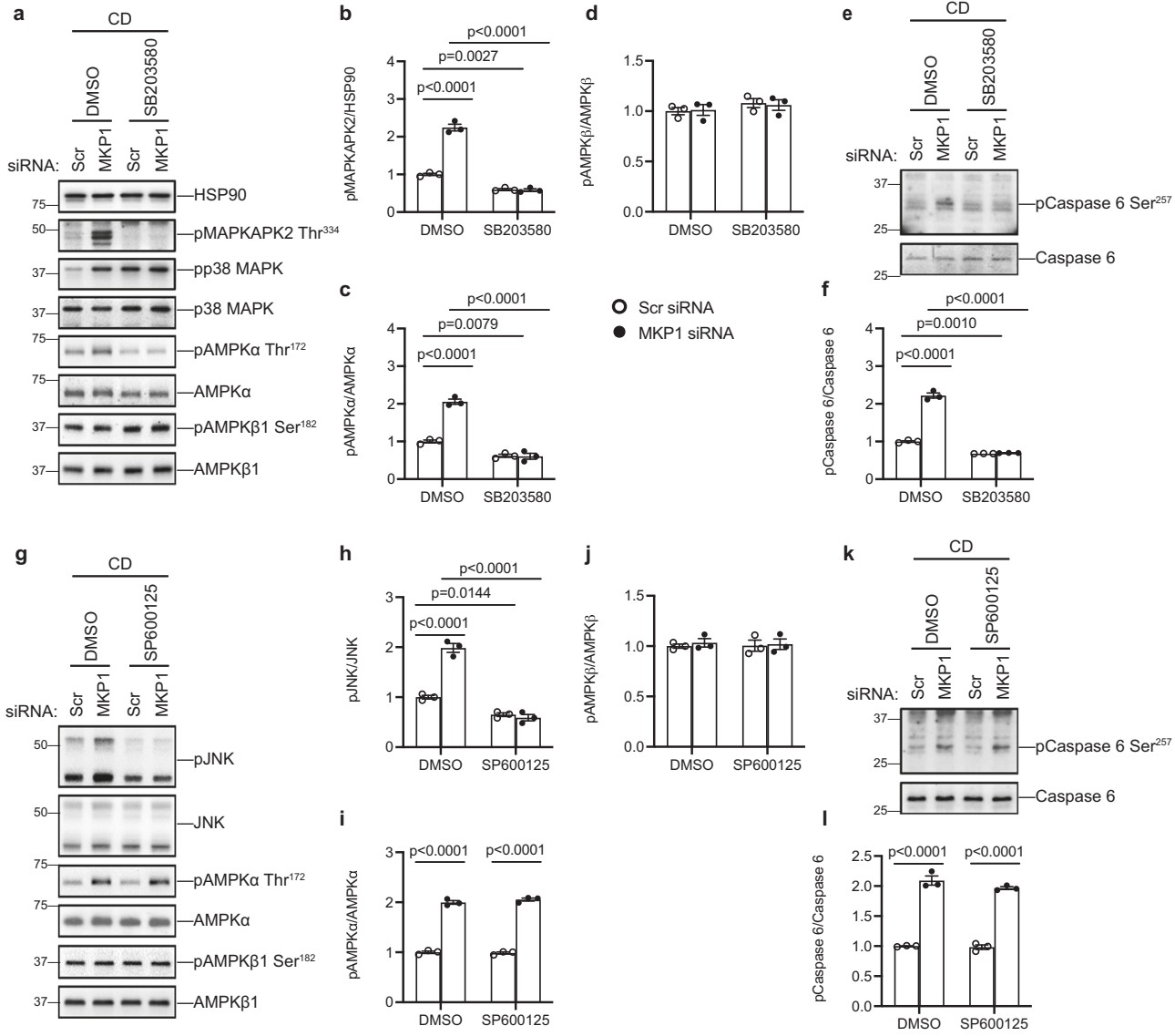

**Fig. 7 | MKP1-p38 MAPK signaling is required for AMPKα-mediated caspase 6 phosphorylation.** HepG2 cells were transfected with scrambled siRNA (Scr) or MKP1 siRNA for 24 h followed by 24 h starvation in serum-free medium. Cells were treated with CD medium in the presence of DMSO, 10 μM SB203580 or 10 μM SP600125 for 4 h (for detecting MAPKs and AMPK pathway in **a** and **g**) or 24 h (for detecting phosphorylation of caspase 6 in **e** and **k**). **a** Cells were treated with SB203580. Immunoblots of phospho-MAPKAPK2 (Thr334) and HSP90 as a loading control or phospho-p38 MAPK, phospho-AMPKα (Thr172), phospho-AMPKβ1 (Ser182) with the indicated corresponding totals. **b**–**d** Quantitation of immunoblots

from (**a**). **e** Immunoblots of phospho-caspase 6 (Ser257) and caspase 6.
**f** Quantitation of immunoblots from (**e**). **g** Cells were treated with SP600125. Immunoblots of phospho-JNK, phospho-AMPKα (Thr172), phospho-AMPKβ1 (Ser182) with the indicated corresponding totals. **h**–**j** Quantitation of immunoblots from (**g**). **k** Immunoblots of phospho-Caspase 6 (Ser257) and Caspase 6.
**l** Quantitation of immunoblots from (**k**). Key: CD choline-deficient medium. Data represent the mean ± SEM from 3 independent experiments. *p*-values were determined by two-way ANOVA.

formally determined whether the MAPKs are involved in the progression of NASH. Here we show that the JNK and p38 MAPK phosphatase, MKP1, is induced by oxidative stress and via p38 MAPK dephosphorylation represses AMPKα activity leading to activation of the caspase-6/3 apoptosis pathway. MKP1 exerts its negative effect on AMPKα by restricting LKB1 nuclear exit thus limiting access to AMPKα. The initiation of hepatocellular death by this mechanism provides an additional link between increased oxidative stress seen in metabolic syndrome and the suppression of AMPKα which facilitates pro-inflammatory responses that promote hepatic fibrosis (Supplemental Fig. 9).

NAFL is characterized by an accumulation of lipid within the hepatocyte[1,66]. The overload of intracellular lipids in hepatocytes induces an acute activation of pathways such as the unfolded protein

response, oxidative stress, and pro-inflammatory signals that promote cellular injury and death[67]. MKP1 is an immediate-early gene that is upregulated in response to oxidative stress[24]. MKP1 expression was found to be increased in obese humans, in skeletal muscle, mononuclear cells and in subcutaneous adipocyte tissues[32,33]. We have shown that MKP1 is upregulated in the liver of HFD-fed mice and liver-specific MKP1 deleted mice are resistant to HFD-induced NAFL[26]. In this study, we demonstrated that MKP1 expression was upregulated in the livers of human patients who are either obese with steatosis or are obese with NASH relative to obese patients with no steatosis. These observations are in line with those where it was reported that SNPs associated with MKP1 negatively correlate with metabolic syndrome in obese patients[31]. It would be important to complement these studies with further analyses of MKP1 protein expression in liver sections

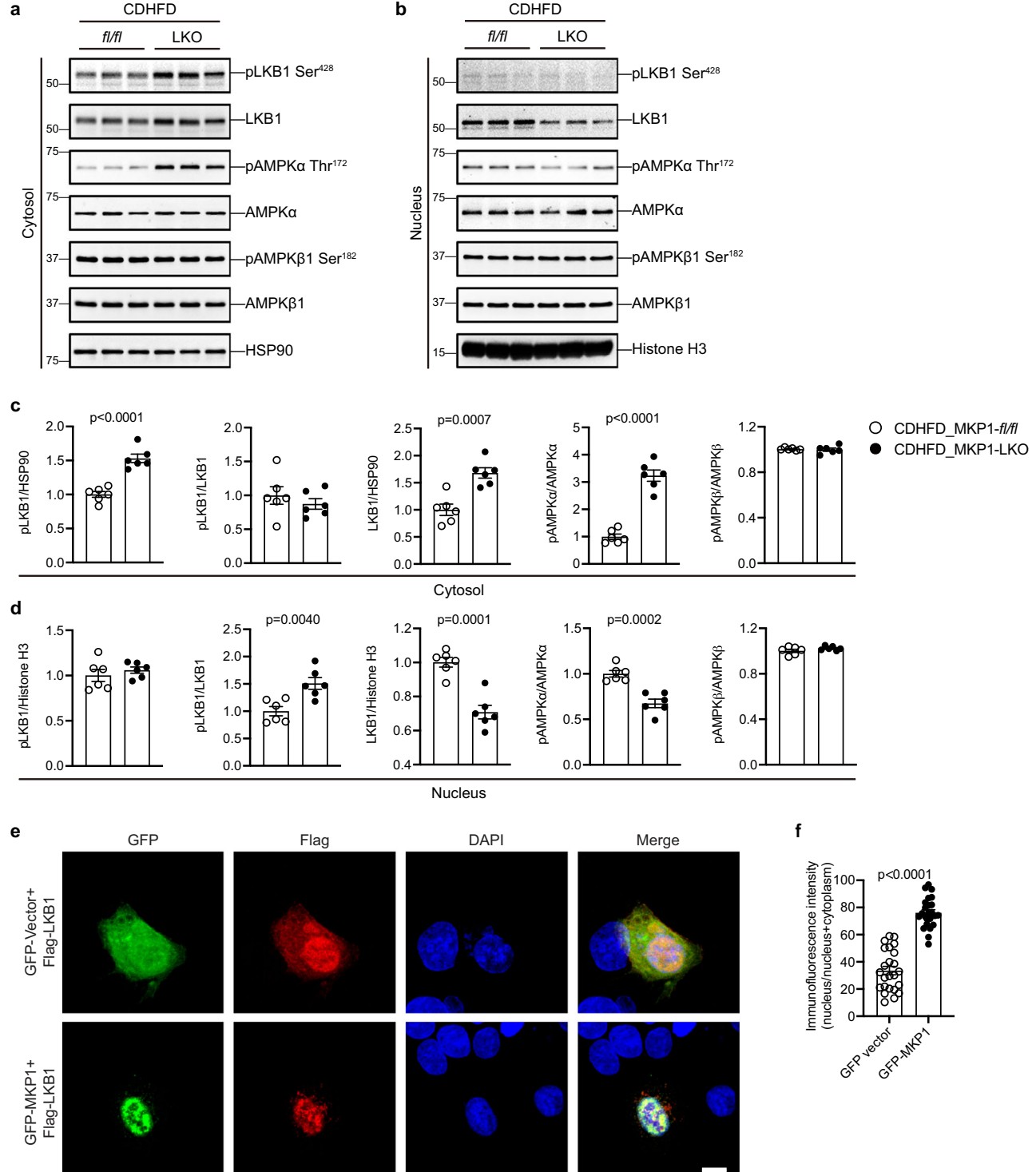

**Fig. 8 | MKP1 regulates LKB1 Ser428 phosphorylation and nuclear localization.**
**a**, **b** $Mkp1^{fl/fl}$ and MKP1-LKO mice were fed with either a chow or CDHFD diet for 12 weeks. Cytoplasmic and nuclear protein fractions were enriched from livers. Immunoblots of pLKB1 (Ser428) and LKB1, pAMPKα (Thr172) and AMPKα in (**a**) cytosol and (**b**) nucleus. Key: $fl/fl$, $Mkp1^{fl/fl}$ mice; LKO, MKP1-LKO mice. **c**, **d** Densitometry of corresponding immunoblots from (**a**, **b**). Data represents mean ± SEM of 6 mice per group. *p*-value was determined by two-sided student's

unpaired *t*-test. **e** HepG2 cells were transfected with GFP vector or GFP-MKP1 with Flag-LKB1 for 24 h. Cells were stained with anti-Flag antibody and corresponding Alexa-594 secondary antibody. Scale bar = 10 μm. **f** Quantification of distribution of Flag-LKB1 signal as a ratio of nucleus/nucleus + cytosol. Data were collected from 24 cells in each group across 3 independent experiments. Data represent the mean ± SEM from 3 independent experiments. *p*-value was determined by two-sided student's unpaired *t*-test.

derived from human NASH patients. We modeled human NASH using both the CDAA and CDHFD diets (NASH diets) and in each case hepatocyte-specific deletion of MKP1 curtailed the development of NASH. The protection from NASH is likely due to the deletion of

hepatic MKP1 because of the observation that isolated hepatocytes from NASH fed mice exhibited increased MKP1 protein expression and reduced p38 MAPK and JNK activities in wild type animals that progressed to NASH. Conversely, hepatocytes isolated from MKP1-LKO

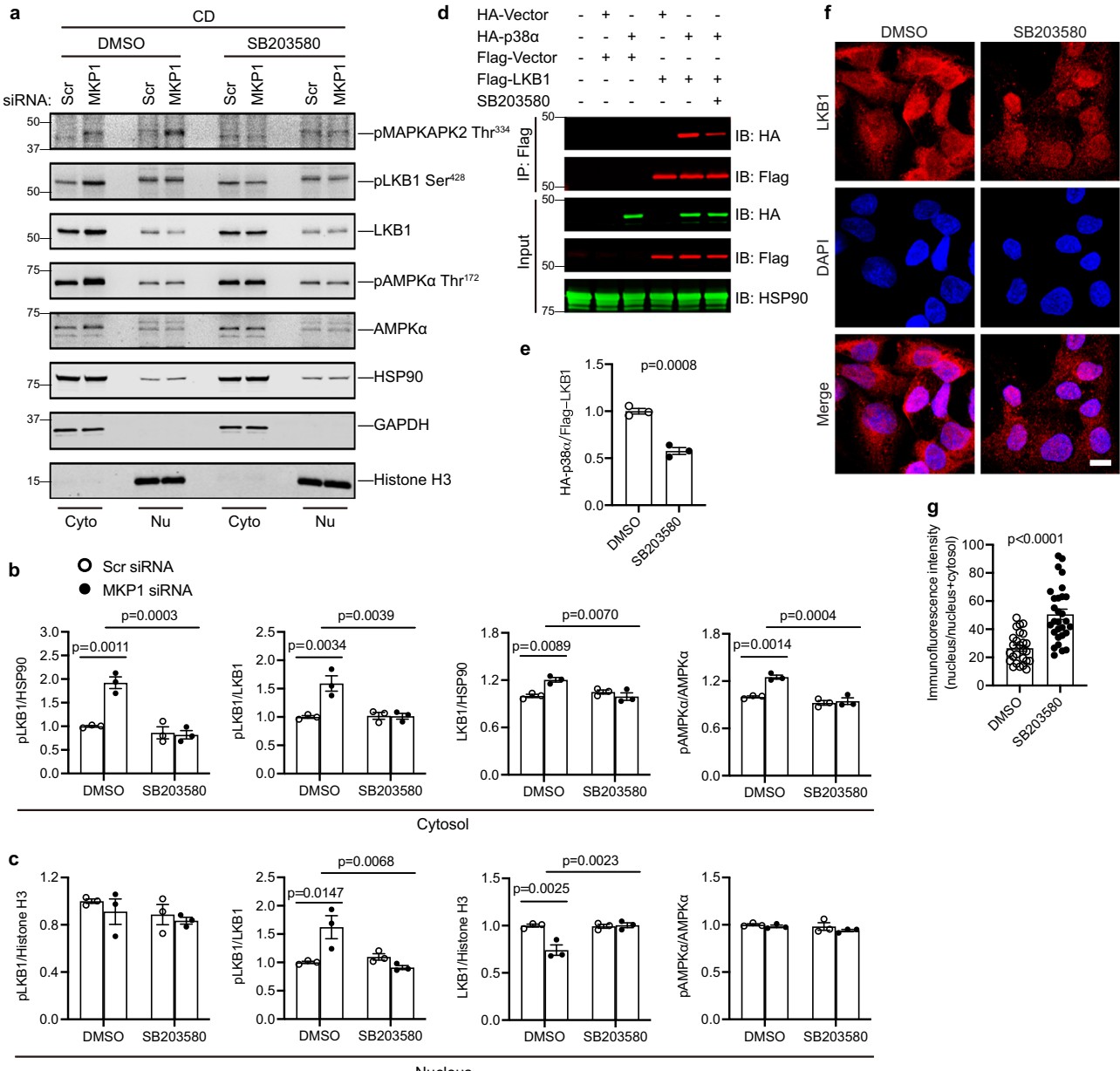

**Fig. 9 | p38 MAPK activity is required for p38 MAPK/LKB1 interaction and LKB1 nuclear exit. a** HepG2 cells were transfected with scrambled siRNA (Scr) or MKP1 siRNA for 24 h followed by 24 h starvation in serum-free medium. Cells were treated with CD medium in the presence of DMSO or 10 μM SB203580 for 4 h. Cytoplasmic and nuclear fractions were enriched and the expression of pLKB1 (Ser428) and LKB1, pAMPKα (Thr172) and AMPKα in the cytosol and nucleus were determined by immunoblotting. **b, c** Densitometry of corresponding immunoblots from (**a**). Key: Scr scrambled siRNA, CD choline-deficient medium, Cyto cytosol, Nu nucleus. Data represent the mean ± SEM from 3 independent experiments. *p*-values in (**b**) and (**c**) were determined by two-way ANOVA. **d** HepG2 cells were transfected with the indicated expression plasmids for 24 h. Cells were treated with DMSO or 10 μM SB203580 for 24 h. Cell lysates were harvested and used for immunoprecipitation with an anti-Flag antibody. Flag-biotin, HA-biotin and HSP90 antibodies were used to detect protein. Key: IP immunoprecipitation, IB immunoblotting. **e** Quantification of HA-p38α MAPK levels present in HA-LKB1 immunocomplexes either in the presence of DMSO or 10 μM SB203580 from (**d**). Data represent the mean ± SEM from 3 independent experiments. *p*-value was determined by two-sided student's unpaired *t*-test. **f** HepG2 cells were starved with DMEM overnight followed by PBS starvation for 1 h in the presence of DMSO or 10 μM SB203580. Cells were fixed and stained with LKB1 antibody. Scale bar = 10 μm. **g** Quantification of LKB1 fluorescence distribution as a ratio of nucleus / nucleus + cytosol. Data were collected from 26 cells in the DMSO group and 29 cells in the SB203580 group across 3 independent experiments. The results are represented as the mean ± SEM from these 3 independent experiments. *p*-value was determined by Mann–Whitney test.

mice fed a NASH diet that were protected from the development of NASH exhibited enhanced p38 MAPK and JNK activities. The loss of hepatocyte MKP1 increased the nuclear pool of both p38 MAPK and JNK activities in NASH diet fed animals. We have shown previously that MKP1 negatively regulates fatty acid oxidation through a PPARα-dependent pathway[30] and promotes de novo lipogenesis via PPARγ[28]. Thus, the early lipid accumulation that is evoked in NASH fed animals is repressed in the absence of MKP1 due, at least in part, to the combined effects of increased fatty acid oxidation[30] and decreased de novo lipogenesis[28] which collectively maintains hepatic oxidative homeostasis and hepatocellular viability. Dysregulation of redox balance as a result of unbridled lipid accumulation as seen in NASH fed animals leads to lipotoxic stress and the generation of $H_2O_2$[37,50,67] which induces MKP1 expression and causes local nuclear dephosphorylation of p38 MAPK and JNK. Altered mitochondrial function which occurs as a consequence of mitochondrial membrane damage may serve as one

axis of chronic redox impairment that maintains the increased expression of MKP1. Indeed, increased MKP1 expression persisted from patients with simple steatosis to those with NASH. Presumably, chronic hepatic oxidative stress is required to maintain elevated MKP1 levels and this chronic upregulation disrupts the nuclear pool of hepatic MAPK activities that ultimately triggers hepatocellular death. CSAA diet fed mice exhibited weight gain, hepatic steatosis but did not progress to fibrosis. Whereas CDAA diet fed mice exhibit similar characteristics in addition to hepatocellular death, inflammation and fibrosis. Therefore, the CSAA and CDAA diets differ at the level of hepatocellular death implying that the accumulation of hepatic lipid by itself in this model is not sufficient to drive fibrosis. Since MKP1 is upregulated in CDAA diet fed mice it implies that the CDAA diet evokes fibrosis in a manner that correlates with the upregulation of MKP1 and hepatocellular death rather than just hepatic steatosis, consistent with the notion that multiple "hits" are required for the development of NASH.

Reduced activity of AMPK had been associated with the onset of hepatic fatty acid accumulation and NAFL/NASH[41,42]. However, deletion of AMPK neither causes NAFL nor promotes NASH. Instead, re-activation or constitutive activation of AMPK is sufficient to prevent the development of NAFL and NASH[43]. Moreover, patients with type II diabetes have been reported to have lower ATP levels despite being in net nutrient excess[45]. Provocatively, the reduction of AMPK does not appear linked to a switch in energy homeostasis. How then does AMPK decline in its activity in NASH in a manner seemingly independent of nutritional status? LKB1, the major upstream kinase that is responsible for the activation of AMPKα, is localized in the nucleus and becomes active upon binding to STRADα and MO25α when exported to the cytoplasm[47,48]. We revealed increased cytoplasmic LKB1 expression with concomitant decreased nuclear expression levels in MKP1-LKO mice fed a NASH diet as well as in experiments where MKP1 was knocked down in HepG2 cells. Moreover, there was increased nuclear LKB1 when MKP1 was overexpressed in HepG2 cells. Collectively, these data indicate that MKP1 inhibits the nuclear exit of LKB1. By using inhibitors of p38 MAPK and JNK, we revealed that p38 MAPK mediated the regulation of MKP1 on AMPKα, but not JNK. Furthermore, a complex between p38 MAPK and LKB1 was detected, with this interaction being dependent upon p38 MAPK activity. These results suggest that inactivation of p38 MAPK by MKP1 disassociates LKB1 from p38 MAPK leading to reduced phosphorylation and nuclear retention. Furthermore, inhibition of p38 MAPK activity impaired LKB1 nuclear exit. Consistent with our results, mice fed a HFD have been shown to exhibit reduced p38 MAPK following HFD-induced obesity and the liver-specific p38 MAPK knockout mice exhibit exacerbated steatosis when fed a HFD indicating that p38 MAPK protects against HFD-induced NAFL[18,19]. These data support the notion that the decline in nuclear p38 MAPK activity as a result of MKP1 upregulation in NASH fed mice inhibits AMPKα activity by retaining the localization of LKB1 to the nucleus.

Although there has been much work on the actions of JNK and p38 MAPK in the development of NAFL[8,9,14–17] the role of hepatic MAPKs in promoting the essential pro-inflammatory "hit" required to drive fibrosis in the transition from NAFL to NASH has not been formally demonstrated. To the extent that JNK1/2 inhibition in the liver appears to protect from hepatocellular death and fatty liver disease[8,68], our data point to a MKP1-p38 MAPK-LKB1 axis that is spatially restricted to the nucleus that promotes hepatocellular death. JNK does not appear to regulate LKB1 phosphorylation, and hence localization, consistent with the failure for JNK inhibition in the MKP1-deficient background to rescue hepatocellular death. Evidence for p38 MAPK at least in the development of NAFL has been previously provided, where inactivation of hepatocyte p38 MAPK promotes NAFL by activating the UPR pathway[19]. Invariably, these pathways cooperate and thus, how MKP1 affects the p38 MAPK-UPR pathway would be important to assess in

the NAFL to NASH transition[21]. To the best of our knowledge our data assigns hepatic MAPKs specifically to the development of NASH, without contradicting the involvement of the MAPKs in the development of NAFL[8,9,14–17]. It is important to note that hepatocyte-specific MAPK-deficient mouse models do not elicit spatially restricted substrate phosphorylation as we have shown here. Until the repertoire of the MAPK liver-specific deficient mouse models is tested for their effects when fed NASH diets comparisons to the phenotypes described herein with MKP1 will be premature.

Hepatocyte death represents a fundamental mechanism of NASH pathogenesis as this event represents an essential driver of the inflammatory process that defines NASH[3,5]. The insult to the hepatocyte arising through stresses such as lipotoxicity manifests histological phenotypes of injury such as hepatocyte ballooning which is linked to the severity of NASH[69]. We demonstrate that NASH-fed MKP1-LKO mice are resistant to the development of hepatocyte ballooning and ultimately hepatocyte death. Mechanistically, we found that hepatocyte death occurs through the activation of an AMPK-caspase 6 pathway whereby AMPK phosphorylates caspase 6 at a site that negatively regulates the ability of caspase 6 to be cleaved by caspase 3[42]. Autoprocessing of caspase 6 is not sufficient to initiate apoptosis but rather it represents a feed-forward loop that sustains the apoptotic signal[41,42]. In this regard, the ability of MKP1 when overexpressed to promote this pathway suggests that its actions persist to later stages of NASH pathogenesis as well as the earlier stages where it is induced as a result of oxidative stress. This inference is supported by the upregulation of MKP1 observed in human patients that are both obese with steatosis and obese with NASH. Persistent hepatocyte death mediated through MKP1 overexpression promotes apoptosis, or possibly through other mechanisms such as either necroptosis or pyroptosis, which we cannot exclude, leads to the recruitment of inflammatory cells that drive the fibrotic response.

The hepatic stellate cell (HSC) represents the major cell type of the fibrotic response[70,71]. HSCs are activated by cytokines derived from inflammatory cells that are recruited to the site of hepatocellular injury[70,71]. We found that NASH fed MKP1-LKO mice are absent of marked increased levels of infiltration of inflammatory cells consistent with there being limited hepatocellular damage in these mice. Indeed, we observed decreased inflammation and reduced fibrosis in NASH-fed MKP1-LKO mice compared to *Mkp1*[fl/fl] mice. Previous studies have reported that whole body MKP1-deficient mice displayed biochemical and histological evidence of more severe hepatocellular injury and inflammation induced by either acetaminophen[72] or carbon tetrachloride (CCl$_4$)[73,74]. In this study the diet-induced NASH model is chronic in contrast to the xenobiotic insult of acute liver injury. The reconciling explanation between these observations and those reported here is likely related to the initiating insult that is distinct between a xenobiotic overload which is acute as compared with a chronic accumulation of lipids that eventually manifest as lipotoxic. Ostensibly, mitochondrial substrate overload will be a driving nidus for nutritional excess that is seen in obesity rather than acute toxic insult. Thus, the mechanism of toxicity is a likely differentiator although it is still needs to be fully established whether patients with xenobiotic-induced injury are more susceptible to the development of NASH[75].

Although we propose a model whereby MKP1 overexpression promotes the inactivation of AMPK other modes of MKP1 regulation in the AMPK pathway have been reported. MKP1 has been reported to be phosphorylated at Ser334 by AMPKα2, which results in MKP1 ubiquitination-dependent degradation in adipocytes as well as isolated hepatocytes[49]. Under conditions in which animals are fed a NASH diet, MKP1 expression is upregulated when AMPKα activity is reduced. It is possible that reduced AMPK activity in NASH fed animals leads to reduced Ser334 phosphorylation that facilitates the upregulation of MKP1 post-translationally in addition to its upregulation transcriptionally through increased oxidative stress. Since ROS also

depresses AMPK activity this could lead to both transcriptional upregulation of MKP1 and reduced degradation through AMPK. Conversely, excessive ROS has been reported to directly activate AMPK through S-glutathionylation of cysteines on the AMPKα subunit[76]. The activation of AMPKα upon MKP1 downregulation supports the interpretation that MKP1 regulates LKB1. We found no difference in the phosphorylation status of AMPKβ at Ser182 consistent with the notion that this site is neither phosphorylated by LKB1 nor does it regulate AMPK activity[77,78]. Further, AMPKβ Ser182 phosphorylation has been reported to be constitutive in the liver[77]. In contrast, the activity of AMPKα is suppressed in diet-induced obesity and NASH[41,42]. Consistent with previous studies a depressed level of AMPKα activity was found in CDAA diet-fed *Mkp1*[fl/fl] mice as well as in choline deficient treated HepG2 cells. Mechanistically, suppression of AMPKα activity under NASH conditions relieved the inhibition for caspase 6 cleavage, which facilitated hepatocyte apoptosis and promoted NASH[42]. Consistent with this, *Mkp1*[fl/fl] mice fed a CDAA diet exhibited decreased levels of phospho-caspase 6 and a concomitant increase in the expression of the cleaved caspase 6 form. Liver-specific AMPKα activation protects against diet-induced obesity, hepatic steatosis and liver fibrosis[43,44]. An increased activation of AMPKα activity was detected in CDAA diet-fed MKP1-LKO liver or choline deficient treated MKP1 knockdown HepG2 cells, as well as decreased activation of caspase 6. Further, we revealed that compound C, an AMPKα inhibitor or AMPK knockdown, reversed the above effects of MKP1 deficiency. The precise mechanism of how MKP1-p38 MAPK regulates LKB1 sub-cellular localization is attributed to the ability of MKP1 to regulate Ser428 which has been shown to promote LKB1 nuclear exit in a p38 MAPK-dependent manner. We showed that p38 MAPK activity promotes its interaction with LKB1 suggesting a mechanism whereby increased MKP1 expression would lead to reduced p38 MAPK activity and subsequently reduced p38 MAPK binding to LKB1. It remains to be determined whether p38 MAPK directly phosphorylates LKB1 at Ser428, this site is not within the consensus typically phosphorylated by the MAPKs, and so alternative mechanisms of p38 MAPK regulation of Ser428 on LKB1 are likely operative. In addition to controlling the sub-cellular localization of LKB1 it has also been reported that Ser428 phosphorylation on LKB1 promotes its activity[64]. However, it is not clear whether the increased activity measured on LKB1 as a result of Ser428 phosphorylation occurs as an indirect consequence of LKB1 moving from the nucleus to the cytosol where it then interacts with MO25 and STRAD[64]. Nevertheless, our data collectively demonstrate that MKP1/p38 MAPK coordinates LKB1 Ser428 phosphorylation to regulate its nuclear localization and subsequent access to AMPKα (Supplemental Fig. 9). This model also suggests that downregulation of AMPKα during NAFL/NASH is dependent upon LKB1 restriction to the nucleus to the extent that nutritional sensitivity of AMPKα may no longer be rate-limiting to its activity.

Herein we have uncovered a mechanism of AMPK depression in the NAFL to NASH transition, which is mediated by a nuclear MKP1/p38 MAPK-LKB1 axis that promotes hepatocyte death. These results demonstrate a role for MKP1/p38 MAPK in the development of NASH transition. We have identified MKP1 as a target that promotes hepatocyte death which is a prerequisite to transit to fibrotic disease. Given the recent developments that have revealed an allosteric site exists on the MKPs that render them druggable[79–81], these data raise the possibility that MKP1 may represent a potential therapeutic target to curtail the progression of NASH.

## Methods

### Human liver biopsies

The use of human tissue was approved by the Monash University Human Research Ethics Committee (CF12/2339-2012001246; CF15/3041-2015001282). All subjects gave their written consent before participating in this study. Liver core biopsies were from obese men and women undergoing bariatric surgery have been described previously[52] and were processed for RNA isolation. Gender differences analyses were not performed due to the low frequency of suitable donors.

### Mouse models and isolated hepatocyte isolation

The Yale University School of Medicine Institutional Animal Care and Use Committee approved all animal studies (Protocol #2020-10142). The Generation of *Mkp1*[flox/flox] (*Mkp1*[fl/fl]) and Alb-Cre-*Mkp1*[fl/fl] (MKP1-LKO) mice have been described previously[26]. Male *Mkp1*[fl/fl] and MKP1-LKO mice at 8 weeks old were fed with chow (#2018S, Envigo), choline-sufficient L-amino acid-defined (CSAA) or choline-deficient L-amino acid-defined (CDAA) diet (#518754 and #518753, Dyets Inc.) for 22 weeks, or fed with chow or choline-deficient L-amino acid-defined high fat diet (CDHFD; 35% fat/60% kcal fat; #A06071302, Research Diet) for 12 weeks[42,61]. Mice were housed within a temperature of ~74 ºF (humidity ~44%) with 12/12 h dark/light cycle and given free access to food and water. Primary hepatocyte isolation was performed as previously described[26]. Primary hepatocytes were isolated by collagenase/DNAse perfusion of mouse liver and purified by 45% percoll gradient centrifugation. Cells were then washed, and protein harvested for immunoblotting.

### Cell culture, transfection, and treatments

HepG2 cells (#HB-8065, ATCC) were cultured as described previously[26]. Briefly, the cells were maintained in DMEM (#11965-092, Gibco) contained with 10% fetal bovine serum (#F0926-500ML, Sigma-Aldrich), penicillin streptomycin (#15140-122, Gibco) and sodium pyruvate (#11360-070, Gibco). For siRNA transfection, cells were transfected with scrambled (#4390843, Thermo Fisher Scientific) or MKP1 (#4390824_siRNA ID#s4363, Thermo Fisher Scientific) siRNA at 10 nM final concentration, or scrambled (#AM4611, Thermo Fisher Scientific), AMPKα1 (#AM51331_siRNA ID#767, Thermo Fisher Scientific), AMPKα2 (#AM51331_siRNA ID#772, Thermo Fisher Scientific) siRNA at 30 nM final concentration using DharmaFECT 4 siRNA Transfection Reagent (#T-2004-02, Dharmacon) at a ratio of 1:5 (siRNA: DharmaFECT 4 = μl:μl) according to the manufacturer's instruction. For plasmid transfections, cells were transfected with the indicated plasmids using FuGENE® HD Transfection Reagent (#E2311, Promega Corporation) at a ratio of 1:5 (plasmid: FuGENE® HD Transfection Reagent = μg:μl) according to the manufacturer's recommendations. For choline-deficient (CD) treatment, the cells were starved with DMEM/Ham's F-12 1:1 in a mixture of choline-sufficient (CS) medium (#DFL13-500ML, Caisson Labs) for 24 h, followed by either CS or CD medium (#DFL25-500ML, Caisson Labs) for 4 h or 24 h. For the application of inhibitors, 10 μM Compound C (CC, AMPKα inhibitor; #171261, Sigma-Aldrich), 10 μM SB203580 (p38 inhibitor; #5633, CST), 10 μM SP600125 (JNK inhibitor; #8177, CST), 2 mM N-acetyl-l-cysteine (NAC, antioxidant; #A9165 Sigma-Aldrich) or 20 μM MitoTEMPO (mitochondria-targeted antioxidant; #SML0737, Sigma-Aldrich) were added in the medium 1 h before CD treatment.

### Body composition analysis

The body composition of mice was determined by Echo MRI (Bruker mini-spec Analyzer; Echo Medical Systems, Houston, TX) as previously described[26]. Conscious male *Mkp1*[fl/fl] and MKP1-LKO mice fed for 22 weeks either with CSAA or CDAA diet were subjected to MRI analyses to derive whole body fat and lean mass.

### Histological, immunofluorescence staining and liver chemistries

The mice were euthanized by $CO_2$ inhalation and perfused transcardially with ice-cold phosphate-buffered saline (PBS, pH 7.4). For histological analyses, the liver pieces were fixed with 4% paraformaldehyde (PFA) overnight followed by gradual ethanol dehydration and paraffin embedding. The livers were sectioned at a thickness of 4 μm. Hematoxylin and eosin (H&E) staining and Sirius red

staining were conducted as previously described[26,82]. Following deparaffinization and rehydration, sections were subjected to staining using either hematoxylin and eosin or Sirius red, after which they underwent washing, dehydration, clearing, and mounting. Stained tissue sections were visualized under a bright-field microscope equipped with a digital camera (BX51, Olympus©). To prepare frozen sections, liver pieces were fixed with 4% PFA followed by dehydration overnight in 20% sucrose and were embedded in OCT and sectioned at 10 μm thickness as described[26]. Oil Red O staining was performed as previously described[26]. After air drying, cryosections were stained with Oil Red O solution, followed by washing and mounting. Stained tissue sections were visualized under a bright-field microscope equipped with a digital camera (BX51, Olympus©). TUNEL staining was performed according to the manufacturer's instruction (#11684795910, Roche). Liver slides or cells were fixed with 4% PFA and permeabilized with 0.1% triton-X-100 in 0.1% sodium citrate. After being incubated with TUNEL reaction mixture for 60 min at 37 °C in the dark, the slides or cells were mounted with VECTASHIELD® HardSet™ Antifade Mounting Medium (#H-1400-10, Vector Laboratories). For immunofluorescence labeling, the liver tissue slides or cells on the coverslips were fixed, permeabilized, blocked and incubated with primary and secondary antibodies. The nucleus was stained with VECTASHIELD® HardSet™ Antifade Mounting Medium. The immunofluorescence signal was captured by confocal microscopy with Zeiss Zen software (ZEISS). For the quantification of liver sections, statistical data were acquired from 5 randomly selected fields for each mouse. In cell-based experiments, the number of selected fields or cells were indicated in the figure legend. The antibodies used in this section are listed in Supplemental Table 2. Hepatic TG content was measured as described previously[26]. Frozen liver tissue (100 mg) was added with 1 mL chloroform:methanol (1:1), homogenized and rotated for 30 min. After adding 200 uL of 1 M $H_2SO_4$, the homogenized samples were centrifuged. Then the organic layer containing triglyceride was collected and the concentration was measured by triglyceride reagent (#236-60, Sekisui) with a standard (#SE-035, Sekisui).

### RNA extraction and real-time PCR

Total RNA was extracted using RNeasy kit according to the manufacturer's instruction (#74104, Qiagen). The amount of 1 μg of total RNA was used to synthesize cDNA by using High-Capacity cDNA Reverse Transcription kit according to the manufacturer's instruction (#4368814, Applied Biosystems, Thermo Fisher Scientific). Quantitative real-time PCR was performed using TaqMan Gene Expression Master Mix (#4369016, Applied Biosystems, Thermo Fisher Scientific) and Quanta studio 3 (Applied Biosystems, Thermo Fisher Scientific). The 18 S was used as a housekeeping gene, and expression levels were presented as fold change relative to control. All primers were from Thermo Fisher Scientific and are listed in Supplemental Table 1.

### Protein isolation

For whole cell lysates, tissues or cells were homogenized with RIPA buffer (25 mM Tris-pH7.4, 150 mM NaCl, 5 mM EDTA, 1% NP-40, 0.5% Sodium deoxycholate, 0.1% SDS) containing protease inhibitors (1 mM benzamidine, 1 μg/ml pepstatin A, 5 μg/ml leupeptin, 5 μg/ml aprotinin, 1 mM PMSF) and phosphatase inhibitors (5 mM sodium fluoride (NaF), 1 mM sodium orthovanadate ($Na_3VO_4$)). Lysates were collected by centrifugation and prepared with SDS sample buffer after normalizing protein concentration by BCA reagent. For enrichment of cytoplasmic and nuclear fractions, mouse liver tissue or HepG2 cells were separated with NE-PER Nuclear and Cytoplasmic Extraction Reagents (#78833, Thermo Fisher Scientific) according to the manufacturers' instructions. liver tissue (1 mg) or cells ($2 \times 10^6$) were homogenized by 1 ml CER I buffer. The lysates were added with CER II buffer followed by centrifugation at $12,000 \times g$ for 5 min. The cytoplasmic fractions in the supernatant were collected and the insoluble material were mixed with NER buffer. After being incubated and vortexed, the nuclear fractions were collected by centrifugation at $12,000 \times g$ for 10 min.

### Immunoblotting and co-Immunoprecipitation assays

Protein samples (40 μg) were separated by SDS-PAGE and immunoblotting was performed as previously described[83]. The membranes underwent overnight incubation at 4 °C with primary antibody, followed by a 1-hour incubation at room temperature with secondary antibody. The signal was detected with SuperSignal™ West Dura Extended Duration Substrate (#34075, Thermo Fisher Scientific) and quantified by ImageJ. All phosphorylated proteins were normalized to their corresponding non-phosphorylated total, while others were normalized to either HSP90 or Histone H3 as a loading control. The antibodies used in this study are listed in Supplemental Table 2. HepG2 cells were transfected with the indicated plasmids for 24 h followed by incubation in the presence of DMSO or 10 μM SB203580 for another 24 h. pcDNA 3.1-HA-p38α MAPK and pcDNA 3.1-Flag-LKB1 plasmids were obtained from Drs. Melanie Cobb and Rubin Shaw, respectively. Forty-eight hours after transfection, the cells were used for co-immunoprecipitation experiments as previously described[84]. Briefly, cells were homogenized with NP40 lysis buffer (25 mM Tris-pH7.4, 150 mM NaCl, 5 mM EDTA, 1% NP-40) with protease and phosphatase inhibitors. Lysates were collected by centrifugation and 500 μg of protein were incubated with 0.5 μl of Flag antibody (#F3165, Sigma-Aldrich) at 4 °C overnight. 15 μl of Protein A Sepharose Fast Flow (#17-1279-02, Cytiva Life Sciences) and 15 μl of Protein G Sepharose 4 Fast Flow (#17-0618-01, Cytiva Life Sciences) were added and incubated for 3 h. After being washed, 50 μl of 2×loading buffer was added. The samples were boiled and then separated by SDS-PAGE. The biotin conjugated primary antibodies, Flag-biotin (#F9219, Sigma-Aldrich) and HA-biotin (#12158167001, Roche) antibody, and IRDye 680RD Streptavidin secondary antibody (#926-68079, LI-COR Biosciences) were used, and the signal was detected by Odyssey DLx Imaging System (LI-COR Biosciences) with Image Studio software (LI-COR Biosciences). For full scan blots, see the Source Data file.

### Lipid content measurement

The lipid accumulation in HepG2 cells was determined using BODIPY™ 493/503, following the manufacturer's instructions (#D3922, Thermo Fisher Scientific). Briefly, HepG2 cells were treated under the indicated conditions. Subsequently, the cells were treated with BODIPY™ 493/503 at a final concentration of 2.5 μM and incubated in the dark at 37 °C for 45 min. Afterward, the cells were fixed with 4% PFA at 4 °C for 20 min in the dark and mounted using VECTASHIELD® HardSet™ Antifade Mounting Medium (#H-1400-10, Vector Laboratories). Cell nuclei were stained with DAPI. The immunofluorescence signal was captured using a confocal microscope with Zeiss Zen software (ZEISS). The mean fluorescence intensity (MFI) was quantified using ImageJ per field, with each field containing between 2 and 9 cells. A total of 12 fields were analyzed for each condition (48 cells for CS treated cells and 51 cells for CD treated cells), and the MFI values were then normalized on a per-cell basis.

### Reactive oxygen species (ROS) assay

The ROS content was measured by BODIPY™ 581/591 C11 (Lipid Peroxidation Sensor) according to the manufacturer's instruction (#D3861, Thermo Fisher Scientific). HepG2 cells were treated as indicated and then the cells were incubated with BODIPY™ 581/591 C11 at a final concentration of 5 μM and incubated for 30 min at 37 °C in the dark. The oxidized fluorescence signals were captured by flow cytometry (LSRII, BD) with an excitation at 488 nm and emission at 510 nm, then the mean fluorescence intensity was calculated by FlowJo.

### Statistical analysis

All the data are presented as the mean ± standard error of the mean (SEM) and statistical analyses performed using GraphPad Prism 9

Version 9.5.1 (GraphPad Software Inc.). All data were first subjected to a normality test. For parametric data, the differences between two groups were compared by unpaired student's $t$-test, or among multiple groups were analyzed by one-way or two-way analysis of variance (ANOVA) combined with Tukey's post-hoc test. For nonparametric data, the differences between two groups were compared by Mann–Whitney test, or among multiple groups were analyzed by Kruskal–Wallis test. The $p \leq 0.05$ was considered statistically significant.

## Reporting summary

Further information on research design is available in the Nature Portfolio Reporting Summary linked to this article.

## Data availability

All data needed to evaluate the conclusions in the paper are present in the paper and/or the Supplementary Materials. Source data are provided as a Source Data file. Source data are provided with this paper.

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

## Acknowledgements

This work was supported by NIH grant R01 DK34989 to A.M.B., a National Health and Medical Research Council (NHMRC) of Australia grant to (2008572). C.F.-H. was supported by R35HL135820. A.L. was supported by a grant from the University of Alabama (#251359). We thank the Silvio O. Conte Digestive Diseases Research Core Center (5P30 DK034989) for the isolation of primary hepatocytes, histological and fluorescence microscopy instrumentation. We thank Yale Flow Cytometry (NCI Cancer Center Support Grant #NIH P30 CA016359) for their assistance with flow cytometry service. We thank Yanhong Deng and Maria Ciarleglio for their statistical analyses support. Schematic Fig. 1b, S7a and S9 were created using BioRender.

## Author contributions

A.M.B. and B.Q. was responsible for the conceptualization and design of the study. A.M.B. and B.Q. wrote the manuscript and interpreted the data with intellectual input and approval from all authors. B.Q. was responsible for the design and execution all of the experiments. A.L. was responsible for the initiation of the study. C.E.X. and T.T. provided material and performed the experiment shown in Fig. 1a. Guidance on histological analyses was provided by M.R. W.B. provided the human NASH biopsy samples. Additional technical assistance with reagent generation and metabolic analyses was performed by L.Z. and J.-S.Y. Intellectual insight, review and editing of the manuscript was performed by T.T., C.F.-H., and X.Y.

## Competing interests

The authors declare no competing interests.
