## [Peer Review File · Nature Communications]

MKP1 promotes nonalcoholic steatohepatitis by suppressing AMPK activity through LKB1 nuclear retentionREVIEWER COMMENTS

Reviewer #1 (Remarks to the Author):

This is a new study by Bin Qiu et al. titled, "MKP1 promotes nonalcoholic steatohepatitis by suppressing AMPK activity through LKB1 nuclear retention" to determine the expression of the nuclear-localized mitogen-activated protein kinase (MAPK) phosphatase-1 (MKP1) in NASH diet fed mice and NASH patients.

ISSUES TO BE SOLVED:

1. The introduction provides a commendable, specific background of the topic that gives the reader an appreciation of the importance of studying the molecular mechanism of MKP1 in NASH. However, to create a more significant introduction, authors must clearly include the aim of the study. Also, abstract section can be improved by adding aim of the study and details about methodology.
2. A schematic diagram showing the scheme of the animal model should be included.
3. I suggest changing the term of macrosteatosis to macrovesicular steatosis.
4. Authors must perform microvesicular steatosis determination/quantification
5. Did the authors perform Masson's trichrome staining for extracellular matrix and compared with Sirius red staining?
6. I recommend equalizing the abbreviation of figures in the results section, (lines 203, 207,260 and 268)
7. As suggested above, a more in-depth discussion of Figures 4E and 4G would be helpful. I consider this an important result for this article, and therefore it merits more argumentation.
8. Adding strengths and limitations of the study would be helpful.
9. In the section of statistical analysis, I suggest specifying normality tests (parametric or nonparametric) before Student's unpaired t test and two-way ANOVA choice.
10. I recommend managing a similar reference format in the number of authors following the accepted Nature's set-up. Some references have a dissimilar number of authors compared to the rest of references.
11. I suggest improving the quality and resolution of Supplemental Figure 8 which refers to the model for the regulation MKP1-AMPK-Caspase 6 pathway in NASH.
12. Suppl Fig 3 panel A in image of CDAA/MKP1 fl/fl histology seems to be at bigger

augmentation than the rest of microphotographs

13. Lipid accumulation needs to be validated in HepG2 after exposition to CD medium (Fig 5).

14. Results and discussion section must include a description and analysis of histological findings considering the different hepatic zones (1, 2 and 3) and the role of alphaAMPK1 in the metabolic functions that predominate in each zone. Different metabolic functions have been described in each particular hepatic zones.

Reviewer #2 (Remarks to the Author):

In the current study, Qiu et al. showed that expression of MKP1 is induced by oxidative stress and could repress AMPK α activity via p38 MAPK dephosphorylation, which led to activation of the caspase-6/3 apoptosis pathway in hepatocytes. Thus, Mkp1 deficiency might improve nonalcoholic steatohepatitis. The results are interesting and meaningful. However, the following concerns should be addressed before the manuscript can be published.

Major concerns:

1. Why the study of Figure S1B used the isolated hepatocytes, but not also mouse liver? Why not to show the expression of fibrotic markers in Figure S1?
2. Was the data of Figure S1A independent from the data of Figure S2A-B? Why not to use wild type mice for the study of Figure S1A? How about to combine the Figure S2A and S2B?
3. For the study of Figure S3A, authors tried to show inflammatory infiltrates by H&E staining? Why not do F4/80 immuno- staining here?
4. For the study of Figure S5B, did the authors measure the level of H₂O₂ after N-acetyl-L-cysteine treatment?
5. Why not did the CO-IP study to show the interaction of MKP1 and LKB1 protein?
6. Hepatic steatosis might result in hepatitis. Mkp1 deficiency can improve hepatic steatosis. So, does Mkp1 deficiency directly improve hepatitis, or Mkp1 deficiency only improves hepatic steatosis? How about the effect of Mkp1 deficiency on chronic carbon tetrachloride treatment?
7. For all the pathway inactivation studies, authors only used inhibitors. They should at least

repeat the study of AMPK α inactivation by AMPK α knock-out or knock-down.

8. For Figure S2D and Figure 1H, the liver weight of MKP1-fl/fl mice and Mkp1-LKO mice fed with CDAA were similar, but the TGs content was much lower in Mkp1-LKO group. Can the author explain?

Minor concerns:

1. Gene name for human should use Italic capital letters, like “MKP1” (line 155); while gene name for mouse used as “Mkp1” (line 242).
2. Improve the writing of the manuscript.
3. The relation of MKP1 and AMPK α could be briefly introduced in the Introduction.
4. Did the authors measure plasma level of AST?
5. Did the authors measure α -SMA and/or TGF β in Figure S1B.

A point-by-point response to each of the reviewers comments follows.

Reviewer #1

1. “...authors must clearly include the aim of the study. Also, abstract section can be improved by adding aim of the study and details about methodology.”

We thank the reviewer for providing guidance on how we can strengthen the clarity and improve upon communicating the significance of this work. In the revised abstract and introduction, we have included a clear statement of the goal of the study, addressing the need to determine the role of MKP1 in NASH development and to uncover the mechanism by which it does so. Regarding the abstract, we acknowledge that it would be improved “*by including...details about the methodology.*” However, due to the word limitation set by the journal's guidelines (150 words), we regret that we were unable to incorporate the methodology into the abstract.

2. “A schematic diagram showing the scheme of the animal model should be included.”

We thank the reviewer for this helpful suggestion and we have now incorporated the schematic diagrams for the diet-induced NASH animal models into the revised version of the manuscript. Specifically, **Fig. 1b** now includes the relevant schematic diagram for CSAA/CDAA diet. We have also included a schematic diagram in **Supplementary Fig. 7a** for Chow/CDHFD diet protocol.

3. “I suggest changing the term of macrosteatosis to macrovesicular steatosis.”

We have taken this into consideration and made the necessary modifications throughout the manuscript. The term "macrosteatosis" has been replaced with "macrovesicular steatosis".

4. “Authors must perform microvesicular steatosis determination/quantification.”

We appreciate the importance of determining and quantifying microvesicular steatosis. We have now performed the determination of microvesicular steatosis in our study and have provided the results in **Supplemental Fig. 2j**.

5. “Did the authors perform Masson's trichrome staining for extracellular matrix and compared with Sirius red staining?”

We appreciate the reviewers insights regarding the use of Masson's trichrome staining and Sirius red staining for assessing extracellular matrix and collagen deposition. Both of these staining methods are commonly employed to assess fibrosis, either individually or in combination. In our study, we utilized Sirius red staining to evaluate collagen matrix deposition, and the results are consistent with the mRNA levels of collagen genes, *Coll1a1* and *Col3a1*, as shown in **Fig. 1k**. The demonstrated correlation between Sirius red staining and collagen gene expression supports the fibrotic changes observed. However, to provide a comprehensive evaluation, we also included the assessment of α -smooth muscle actin (α -SMA) protein levels (**Fig. 1l, m**). Notably, the protein expression of α -SMA, an indicator of activated myofibroblasts that give rise to fibrosis, was found to exhibit expression levels that were substantially diminished in NASH diet-fed MKP1-LKO mice as compared with NASH diet-fed *Mkp1^{fl/fl}* mice (**Fig. 1l, m**). In summary, we have assessed fibrosis comprehensively using multiple approaches and these complementary analyses provide a robust evaluation of fibrotic changes in the liver of the MKP1-LKO mouse model used in our study.

6. “I recommend equalizing the abbreviation of figures in the results section, (lines 203, 207,260 and 268)”.

We have carefully reviewed the manuscript and made the necessary adjustments to ensure that the abbreviation of figures is consistent.

7. *“As suggested above, a more in-depth discussion of Figures 4E and 4G would be helpful. I consider this an important result for this article, and therefore it merits more argumentation.”*

We assume the reviewer is referring to the differential effect of MKP1 between AMPK α (Thr172) as compared with AMPK β (Ser182) phosphorylation. The explanation for this difference is related to the fact that LKB1 is not an AMPK β Ser182 substrate. Additionally, AMPK β (Ser182) phosphorylation does not regulate AMPK activity (Warden et al, Biochemical Journal 2001 Vol. 354 (2), pg. 275) consistent with reports that it is constitutively phosphorylated in the liver at this site (Chen et al, FEBS Letters 1999 Vol. 460 (2), pg 343-348). We expand upon this explanation in the Discussion.

8. *“Adding strengths and limitations of the study would be helpful.”*

Throughout the Discussion we have indicated “strengths and limitations” of our work in the form of raising alternative explanations for points of our data. We highlight these limitations in red text in the Discussion.

9. *“In the section of statistical analysis, I suggest specifying normality tests (parametric or nonparametric) before Student’s unpaired t test and two-way ANOVA choice.”*

We have applied normality tests for our entire data set and indicated this in the revised methods section (see Statistical analysis). Depending upon the outcome of the normality tests (parametric or nonparametric) we have applied the appropriate statistical analysis to the corresponding data and have indicated this in the updated figure legends.

10. *“I recommend managing a similar reference format in the number of authors following the accepted Nature’s set-up. Some references have a dissimilar number of authors compared to the rest of references.”*

We have corrected and updated the reference format using the correct Nature Communications Endnote Style to align with the journals recommended citation format.

11. *“I suggest improving the quality and resolution of Supplemental Figure 8 which refers to the model for the regulation MKP1-AMPK-Caspase 6 pathway in NASH.”*

We have regenerated the schematic diagram of the model for the regulation of the MKP1-AMPK-Caspase 6 pathway in NASH, ensuring a high-quality and high-resolution format (**Supplemental Figure 9**).

12. *“Suppl Fig 3 panel A in image of CDAA/MKP1 fl/fl histology seems to be at bigger augmentation than the rest of microphotographs.”*

We have thoroughly reviewed all of the images, specifically **Supplemental Fig. 3a**, and confirmed that all the panels were acquired at the same magnification. The apparent difference in “augmentation” is attributed to the extensive and severe macrovesicular steatosis induced by the CDAA diet, which results in larger lipid droplets compared to the CSAA diet (**Supplemental Fig. 3a**). It is important to note that our findings align with previous research (Figure 2A in Hepatology, 2013 Feb;57(2):577-89), demonstrating the significant macrovesicular steatosis is induced by the CDAA diet.

13. *“Lipid accumulation needs to be validated in HepG2 after exposition to CD medium (Fig 5).”*

In order to validate the lipid accumulation in HepG2 cells following exposure to CD medium (**Fig. 5**), we employed the BODIPY 493/503 stain and conducted confocal microscopy to visualize the lipid accumulation. Our data clearly demonstrate a significant increase in lipid accumulation in HepG2 cells after exposure to CD medium compared with CS medium treatment. These results have been incorporated into a revised **Supplemental Fig. S5a**, and the methodology in the text has been updated to reflect these additions.

14. *“Results and discussion section must include a description and analysis of histological findings considering the different hepatic zones (1, 2 and 3) and the role of alphaAMPK1 in the metabolic functions that predominate in each zone. Different metabolic functions have been described in each particular hepatic zones.”*

The reviewer makes an excellent point regarding the distinct metabolic functions executed in the different hepatic zones. While we agree that discussion of the different hepatic zones as it pertains to AMPK metabolic functions will be insightful we are very hesitant to comment on this as it will lend itself to over interpretation simply based on the fact that these H&E histological sections do not incorporate supporting analyses of AMPK, MKP1, LKB1 and MAPK expression in zones 1, 2 or 3. In the absence of these data, in our opinion, this would make interpretation of the zonal manifestations of AMPK metabolic functions in the MKP1-LKO model overly speculative. This avenue of investigation should be performed using single cell RNA sequencing technologies coupled with established zonation landmark genes to establish the proper spatial position/layer within the lobule (Halpern et al, Nature, 542: 352-356, 2017). There are more sophisticated techniques such as FIB-SEM imaging in conjunction with deep-learning that can also be applied. Such analyses are well-beyond the scope and focus of the current manuscript. We have provided a comprehensive and quantitative analyses of the NASH phenotype in two distinct dietary models (**Fig. 1-4** and **Supplemental Fig. S1**) both of which firmly support our conclusions on the role of MKP1 in NASH development.

Reviewer #2

1. *“Why the study of Figure S1B used the isolated hepatocytes, but not also mouse liver? Why not to show the expression of fibrotic markers in Figure S1?”*

In this study, we addressed the role of MKP1 in hepatocytes for the development of NASH. To accomplish this, we utilized a mouse model wherein hepatocyte-specific deletion of MKP1 was achieved. Therefore, to focus on the mechanism under investigation, we directly assessed the expression of MKP1 in isolated hepatocytes. We should point out to the reviewer that hepatocytes were isolated from mice following NASH-inducing dietary challenge. Thus, the changes in MKP1 expression and MAPK activation status reflect these *in vivo* effects specifically in hepatocytes. Nevertheless, we do perform experiments in mouse liver where we demonstrate the effects of MKP1-deficiency on MAPK and AMPK signaling (**Fig. 4**) and in context of nuclear/cytosolic compartment re-distribution of activated MAPKs, LKB1 and AMPK (**Fig. 8**).

Regarding the fibrotic markers, it is important to note that fibrogenic genes are primarily expressed by activated hepatic stellate cells (HSCs), rather than hepatocytes. Since our study focuses on the role of MKP1 in hepatocytes, we did not examine the expression of fibrotic markers in HSCs. We do however, show decreased myofibroblast activation in the liver, as determined by measuring the expression of α -smooth muscle actin, in CDAA diet fed MKP1-LKO mice (**Fig. 1l, m**).

2. *“Was the data of Figure S1A independent from the data of Figure S2A-B? Why not to use wild type mice for the study of Figure S1A? How about to combine the Figure S2A and S2B?”*

We do apologize for any confusion, the data presented in **Fig. S1a** and **Fig. S2a, b** are completely separate. **Fig. S1a** was included to demonstrate that the *Mkp1^{fl/fl}* mice developed an obesity phenotype when fed with either a CSAA or CDAA diet as compared with chow-fed *Mkp1^{fl/fl}* mice. Next, to provide a direct comparison between MKP1-LKO mice and *Mkp1^{fl/fl}* mice, we performed a completely separate experiment and generated the body weight data for CSAA diet-fed MKP1-LKO mice and CSAA diet-fed *Mkp1^{fl/fl}* mice, as well as from CDAA diet-fed MKP1-LKO mice and CDAA diet-fed *Mkp1^{fl/fl}* mice, respectively. As shown, **Fig. S2a** and **Fig. S2b** allow for a clear visualization of the body weight trends in these different groups. We appreciate the reviewer's suggestion to combine **Fig. S2a** with **Fig. S2b**. However, due to the extremely similar body weights observed between CSAA diet-fed *Mkp1^{fl/fl}* and MKP1-LKO mice, as well as between CDAA diet-fed *Mkp1^{fl/fl}* and MKP1-LKO mice, it would be challenging to discern all four groups effectively in a combined figure. Therefore, we have elected to present **Fig. S2a** and **Fig. S2b** separately.

3. “For the study of Figure S3A, authors tried to show inflammatory infiltrates by H&E staining? Why not do F4/80 immuno-staining here?”

We appreciate your suggestion to utilize F4/80 immunostaining to visualize inflammatory infiltrates in addition to H&E staining. While F4/80 immunostaining is indeed a useful marker for identifying and quantifying macrophages in the liver, we employed alternative methods to validate our findings. In our study, we incorporated CD45 staining, a pan immune cell marker (**Fig. 3a** and **Fig. S3b**), and CD68 staining, a macrophage-specific marker (**Fig. 3b** and **Fig. S3c**), alongside H&E staining to accurately identify and quantify immune cell infiltrates in the liver (**Fig. S3a**). This approach allowed us to assess the presence of immune cells, specifically macrophages, in the liver inflammatory infiltrates. Moreover, to provide additional quantitative evidence of immune cell infiltration, we analyzed the mRNA levels of CD45 (**Fig. 3c**) and CD68 (**Fig. 3f**) in the livers, and these results were consistent with our CD45 and CD68 immunostaining. While F4/80 immunostaining could have been used, we believe that the combination of H&E staining, CD45 and CD68 staining, and mRNA analysis provides a comprehensive assessment to support our conclusions regarding immune cell infiltration in the liver.

4. “For the study of Figure S5B, did the authors measure the level of H₂O₂ after N-acetyl-L-cysteine treatment?”

In a new set of experiments, we employed BODIPY C11 581/591, a lipid-soluble ratiometric fluorescent indicator of lipid oxidation to measure cellular ROS generation in HepG2 cells. The results, presented in a revised **Fig. S5c, d** demonstrated a significant increase in ROS in CD-treated HepG2 cells compared with CS-treated HepG2 cells. Further, as expected both NAC and MitoTEMPO treatments reduced the CD-induced ROS production. These results support the HepG2 cell experimental model.

5. “Why not did the CO-IP study to show the interaction of MKP1 and LKB1 protein?”

We would like to clarify for the reviewer that MKP1 forms a direct interaction with p38 MAPK though its kinase interaction motif in order to execute its dephosphorylation. When MKP1 dephosphorylates p38 MAPK it dissociates from the complex. Our data show that p38 MAPK activity is required for it to complex with LKB1 (**Fig. 9b**). Thus, under NASH/CD conditions where MKP1 is overexpressed p38 MAPK will be dephosphorylated and dissociated from MKP1. In this state, MKP1 will not be part of the p38 MAPK/LKB1 complex, or at least not at very high levels. Consistent with this supposition, we do not detect MKP1 in the cytosol which would suggest it does not complex with LKB1 when it translocates to this compartment (see **Fig. 8e**). For these reasons, we did not attempt experiments to detect MKP1 in complex with LKB1. We attempt to reiterate this point in the

Discussion, “These results suggest that inactivation of p38 MAPK by MKP1 disassociates LKB1 from p38 MAPK leading to reduced phosphorylation and nuclear retention.”

6. *“Hepatic steatosis might result in hepatitis. Mkp1 deficiency can improve hepatic steatosis. So, does Mkp1 deficiency directly improve hepatitis, or Mkp1 deficiency only improves hepatic steatosis?”*

The reviewer raises an important point regarding the role of hepatic steatosis in the progression to hepatitis/NASH. The development of non-alcoholic fatty liver (NAFL) which is reversible, to the more deleterious irreversible state of NASH, has yet to be fully defined. However, it is clear that simple steatosis by itself is not sufficient to cause NASH but likely involves additional “hits” such as hepatocellular death. Consistent with this notion, despite CSAA-fed mice developing steatosis they do not progress to NASH (**Fig. 1**). Only when mice are fed the CDAA diet is NASH induced concomitant with the induction of hepatocyte death which triggers inflammation (**Fig. 1-3**). In the absence of hepatocyte death inflammation does not occur and NASH fails to develop. Since MKP1 deficiency blocks hepatocyte death (**Fig. 2**) and subsequently ameliorates inflammation (**Fig. 3**) we can infer that MKP1 contributes to the development of NASH. We attempt to reiterate this point in the discussion, “Since MKP1 is upregulated in CDAA diet fed mice it implies that the CDAA diet evokes fibrosis in a manner that correlates with the upregulation of MKP1 and hepatocellular death rather than just hepatic steatosis, consistent with the notion that multiple “hits” are required for the development of NASH.”

“How about the effect of Mkp1 deficiency on chronic carbon tetrachloride treatment?”

In the Discussion of the initial submission we pointed out that “Previous studies have reported that whole body MKP1-deficient mice displayed biochemical and histological evidence of more severe hepatocellular injury and inflammation induced by acetaminophen⁷² or hepatotoxin carbon tetrachloride (CCl₄)^{73,74}.”

7. *“For all the pathway inactivation studies, authors only used inhibitors. They should at least repeat the study of AMPKα inactivation by AMPKα knock-out or knock-down.”*

In response to this comment, we have conducted additional experiments to address the inactivation of the AMPK pathway. Instead of using Compound C to inhibit AMPKα, we performed knockdown experiments using siRNAs to target both AMPKα1 and AMPKα2 simultaneously in HepG2 cells. Consistent with our findings using Compound C, the results from the siRNA knockdown experiments revealed that the effect of MKP1 knockdown on the phosphorylation of AMPKα, as well as the inhibition of caspase 6/3 cleavage, were all blocked upon siRNA knockdown of AMPKα1/2. These new results have been included in **Supplemental Fig. 6**. We appreciate your suggestion, and by incorporating these additional experiments, our study has become more robust in addressing the assignment of MKP1 as an upstream regulator of the AMPK/caspase 6 pathway in NASH development.

8. *“For Figure S2D and Figure 1H, the liver weight of MKP1-fl/fl mice and Mkp1-LKO mice fed with CDAA were similar, but the TGs content was much lower in Mkp1-LKO group. Can the author explain?”*

The reviewer makes an interesting observation about our data. The dissociation from TG content relative to the liver weights between the *Mkp1^{fl/fl}* mice and MKP1-LKO mice fed the CDAA diet suggests that other factors contribute to the overall liver weight under these dietary conditions. As I am sure the reviewer is aware, the liver is constituted with multiple cell types, and changes in hepatocyte lipid content may not necessarily translate to a proportional change in overall liver weight. In addition to lipid accumulation, the liver can store glycogen, which also contributes to liver weight.

Other factors that can affect liver weight include inflammation, fibrosis, and changes in non-hepatocyte cell proliferation. All of these factors may be operative to varying degrees in the MKP1-LKO mice under these dietary conditions. Furthermore, it is possible that the reduced TG content in the MKP1-LKO mice was compensated for by an increase in other metabolites or changes in cellular processes, such as autophagy, which may have contributed to the maintenance of liver weight. It is also noteworthy, that the CDHFD-fed MKP1-LKO mice although protected from the development of hepatic steatosis and NASH did not show significant differences in weight gain. These results suggest that the nature of these NASH inducing diets likely elicit hepatic changes that result in compensatory effects that to varying degrees offset the increase in lipid accumulation. Finally, other researchers have reported that NASH-inducing diets can result in a dissociation between lipid content and liver/body weight – see the following articles. 1) Yang, A., et al., *Hepatology*, (2021) 15, 1122–1135., 2) Yin-hua Ni., et al., *Endocrinology*, (2015) 156 (3), 987–999. PMID: 25562616 and 3) Hakkak R, et al., *Biomed Rep*, (2021) 14 (6):49. PMID: 33859820.

Minor concerns:

1. “Gene name for human should use *Italic capital letters*, like “MKP1” (line 155); while gene name for mouse used as “Mkp1” (line 242)”

We apologize for any confusion caused. In human, the gene name for MKP1 is now correctly represented as "DUSP1" and should be written in italic capital letters as per the standard nomenclature (line 155). On the other hand, in mouse, the gene name for MKP1 is "*Dusp1*" and should be written in italic letters with the initial letter capitalized (line 242). We have now made the necessary corrections throughout the entire manuscript to ensure the proper representation of the gene names in both human and mouse species.

2. “Improve the writing of the manuscript”.

We have made improvements to the writing of the manuscript.

3. “The relation of MKP1 and AMPK α could be briefly introduced in the Introduction”

Thank you for your suggestion to enhance the understanding of our readers regarding the relationship between MKP1 and AMPK α . Based on your suggestion, we have now incorporated a brief introduction to the relationship between MKP1 and AMPK, and highlighted the outstanding issues as it relates to NASH.

4. “Did the authors measure plasma level of AST?”

We did not measure the plasma levels of AST in this study, there is existing evidence supporting the relationship between the CDAA diet and elevated AST levels (*J Clin Transl Hepatol*, 2022 Aug 28; 10(4): 577–588), compared to control CSAA diet. However, we have explored several other key indicators of NASH progression and liver damage such as hepatocellular apoptosis and activation of caspase 3. These findings provide evidence of liver damage in this study. While measuring the plasma level of AST would have provided additional data, it is not essential to the main conclusions drawn from this study.

5. “Did the authors measure α -SMA and/or TGF β in Figure S1B.”

Fig. S1b was generated using primary hepatocytes isolated from *Mkp1^{fl/fl}* and MKP1-LKO mice that were fed with CSAA or CDAA diets for 8 weeks at which point fibrosis has yet to develop. Furthermore, it is important to note that α -SMA, which is a marker of activated hepatic stellate cells (HSCs) (*J Gastroenterol Hepatol*, 2012 Mar; 27(Suppl 2): 65–68), and TGF β , which is expressed in non-parenchymal liver cells including liver sinusoidal endothelial cells (LSEC), granulocytes,

macrophages, and HSCs (Cell Tissue Res, 2012; 347(1): 245–256; Hepatobiliary Surg Nutr, 2014 Dec;3(6):386-406; Cells, 2019 Nov; 8(11): 1419), are not expressed in primary hepatocytes. In **Fig. S1b**, our focus was primarily on assessing whether hepatic MKP1 protein was induced by a NASH diet and evaluating its downstream effects on phosphorylated MAPKs. Therefore, the specific analysis of α -SMA or TGF β was not performed. However, we appreciate your attention to these markers, as they are indeed important in the context of liver fibrosis and NASH. Thus, we evaluated the expression and activation of α -SMA in livers from CSAA or CDAA diet-fed *Mkp1^{fl/fl}* and MKP1-LKO mice, and these findings are reported in **Fig. 11, m**.

REVIEWERS' COMMENTS

Reviewer #1 (Remarks to the Author):

The authors of the manuscript Titled: "MKP1 promotes nonalcoholic steatohepatitis by suppressing AMPK activity through LKB1 nuclear retention" have complied with all my queries raised concerning this manuscript.

Juan Armendariz-Borunda

Reviewer #2 (Remarks to the Author):

The concerns and comments of this reviewer has been well addressed.